# Predicting the Outputs of Finite Networks Trained with Noisy Gradients

## Abstract

A recent line of works studied wide deep neural networks (DNNs) by approximating them as Gaussian Processes (GPs). A DNN trained with gradient flow was shown to map to a GP governed by the Neural Tangent Kernel (NTK), whereas earlier works showed that a DNN with an i.i.d. prior over its weights maps to the so-called Neural Network Gaussian Process (NNGP). Here we consider a DNN training protocol, involving noise, weight decay and finite width, whose outcome corresponds to a certain non-Gaussian stochastic process. An analytical framework is then introduced to analyze this non-Gaussian process, whose deviation from a GP is controlled by the finite width. Our contribution is three-fold: (i) In the infinite width limit, we establish a correspondence between DNNs trained with noisy gradients and the NNGP, not the NTK. (ii) We provide a general analytical form for the finite width correction (FWC) for DNNs with arbitrary activation functions and depth and use it to predict the outputs of empirical finite networks with high accuracy. Analyzing the FWC behavior as a function of $n$, the training set size, we find that it is negligible for both the very small $n$ regime, and, surprisingly, for the large $n$ regime (where the GP error scales as $\mathcal{O}(1/n)$). (iii) We flesh-out algebraically how these FWCs can improve the performance of finite convolutional neural networks (CNNs) relative to their GP counterparts on image classification tasks.

## 1 Introduction

Deep neural networks (DNNs) have been rapidly advancing the state-of-the-art in machine learning, yet a complete analytic theory remains elusive. Recently, several exact results were obtained in the highly over-parameterized regime ($N \rightarrow \infty$ where $N$ denotes the width or number of channels for fully connected networks (FCNs) and convolutional neural networks (CNNs), respectively) (Daniely et al., 2016). This facilitated the derivation of an exact correspondence with Gaussian Processes (GPs) known as the Neural Tangent Kernel (NTK) (Jacot et al., 2018). The latter holds when highly over-parameterized DNNs are trained by gradient flow, namely with vanishing learning rate and involving no stochasticity.

The NTK result has provided the first example of a DNN to GP correspondence valid after end-to-end DNN training. This theoretical breakthrough allows one to think of DNNs as inference problems with underlying GPs (Rasmussen & Williams, 2005). For instance, it provides a quantitative description of the generalization properties (Cohen et al., 2019; Rahaman et al., 2018) and training dynamics (Jacot et al., 2018; Basri et al., 2019) of DNNs. Roughly speaking, highly over-parameterized DNNs generalize well because they have a strong implicit bias to simple functions, and train well because low-error solutions in weight space can be reached by making a small change to the random values of the weights at initialization.

Despite its novelty and importance, the NTK correspondence suffers from a few shortcomings: (a) Its deterministic training is qualitatively different from the stochastic one used in practice, which may lead to poorer performance when combined with a small learning rate (Keskar et al., 2016). (b) It under-performs, often by a large margin, convolutional neural networks (CNNs) trained with SGD (Arora et al., 2019). (c) Deriving explicit finite width corrections (FWCs) is challenging, as it requires solving a set of coupled ODEs (Dyer & Gur-Ari, 2020; Huang & Yau, 2019). Thus, there is a need for an extended theory for end-to-end trained deep networks which is valid for finite width DNNs.

Our contribution is three-fold. First, we prove a correspondence between a DNN trained with noisy gradients and a Stochastic Process (SP) which at $N \to \infty$ tends to the Neural Network Gaussian Process (NNGP) (Lee et al., 2018; Matthews et al., 2018). In these works, the NNGP kernel is determined by the distribution of the DNN weights at initialization which are i.i.d. random variables, whereas in our correspondence the weights are sampled across the stochastic training dynamics, drifting far away from their initial values. We call ours the *NNSP correspondence*, and show that it holds when the training dynamics in output space exhibit ergodicity.

Second, we predict the outputs of trained finite-width DNNs, significantly improving upon the corresponding GP predictions. This is done by deriving leading FWCs which are found to scale with width as $1/N$. The accuracy at which we can predict the empirical DNNs' outputs serves as a strong verification for our aforementioned ergodicity assumption. In the regime where the GP RMSE error scales as $1/n$, we find that the leading FWC are a decaying function of $n$, and thus overall negligible. In the small $n$ regime we find that the FWC is small and grows with $n$. We thus conclude that finite-width corrections are important for intermediate values of $n$ (Fig. 1).

Third, we propose an explanation for why finite CNNs trained on image classification tasks can outperform their infinite-width counterparts, as observed by Novak et al. (2018). The key difference is that in finite CNNs weight sharing is beneficial. Our theory, which accounts for the finite width, quantifies this difference (§4.2).

Overall, the NNSP correspondence provides a rich analytical and numerical framework for exploring the theory of deep learning, unique in its ability to incorporate finite over-parameterization, stochasticity, and depth. We note that there are several factors that make finite SGD-trained DNNs used in practice different from their GP counterparts, e.g. large learning rates, early stopping etc. (Lee et al., 2020). Importantly, our framework quantifies the contribution of finite-width effects to this difference, distilling it from the contribution of these other factors.

## 1.1 RELATED WORK

The idea of leveraging the dynamics of the gradient descent algorithm for approximating Bayesian inference has been considered in various works (Welling & Teh, 2011; Mandt et al., 2017; Teh et al., 2016; Maddox et al., 2019; Ye et al., 2017). However, to the best of our knowledge, a correspondence with a concrete SP or a non-parametric model was not established nor was a comparison made of the DNN's outputs with analytical predictions.

Finite width corrections were studied recently in the context of the NTK correspondence by several authors. Hanin & Nica (2019) study the NTK of finite DNNs, but where the depth scales together with width, whereas we keep the depth fixed. Dyer & Gur-Ari (2020) obtained a finite $N$ correction to the linear integral equation governing the evolution of the predictions on the training set. Our work differs in several aspects: (a) We describe a different correspondence under different a training protocol with qualitatively different behavior. (b) We derive relatively simple formulae for the outputs which become entirely explicit at large $n$. (c) We account for all sources of finite $N$ corrections whereas finite $N$ NTK randomness remained an empirical source of corrections not accounted for by Dyer & Gur-Ari (2020). (d) Our formalism differs considerably: its statistical mechanical nature enables one to import various standard tools for treating randomness, ergodicity breaking, and taking into account non-perturbative effects. (e) We have no smoothness limitation on our activation functions and provide FWCs on a generic data point and not just on the training set.

Another recent paper (Yaida, 2020) studied Bayesian inference with weakly non-Gaussian priors induced by finite-$N$ DNNs. Unlike here, there was no attempt to establish a correspondence with *trained* DNNs. The formulation presented here has the conceptual advantage of representing a distribution over *function space* for arbitrary training and test data, rather than over specific draws of data sets. This is useful for studying the large $n$ behavior of learning curves, where analytical insights into generalization can be gained (Cohen et al., 2019).

A somewhat related line of work studied the mean field regime of shallow NNs (Mei et al., 2018; Chen et al., 2020; Tzen & Raginsky, 2020). We point out the main differences from our work: (a) The NN output is scaled differently with width. (b) In the mean field regime one is interested in the dynamics (finite $t$) of the distribution over the NN parameters in the form of a PDE of the Fokker-Planck type. In contrast, in our framework we are interested in the distribution over function

space at equilibrium, i.e. for $t \to \infty$. (c) It seems that the mean field analysis is tailored for two-layer fully-connected NNs and is hard to generalize to deeper nets or to CNNs. In contrast, our formalism generalizes to deeper fully-connected NNs and to CNNs as well, as we showed in section 4.2.

## 2 THE NNSP CORRESPONDENCE

In this section we show that finite-width DNNs, trained in a specific manner, correspond to Bayesian inference using a non-parametric model which tends to the NNGP as $N \to \infty$. We first give a short review of Langevin dynamics in weight space as described by Neal et al. (2011), Welling & Teh (2011), which we use to generate samples from the posterior over weights. We then shift our perspective and consider the corresponding distribution over functions induced by the DNN, which characterizes the non-parametric model.

**Recap of Langevin-type dynamics** - Consider a DNN trained with *full-batch* gradient descent while injecting white Gaussian noise and including a weight decay term, so that the discrete time dynamics of the weights read

$$\Delta w_t := w_{t+1} - w_t = -\left(\gamma w_t + \nabla_w \mathcal{L}\left(z_w\right)\right) dt + \sqrt{2T dt}\xi_t \tag{1}$$

where $w_t$ is the vector of all network weights at time step $t$, $\gamma$ is the strength of the weight decay, $\mathcal{L}(z_w)$ is the loss as a function of the output $z_w$, $T$ is the temperature (the magnitude of noise), $dt$ is the learning rate and $\xi_t \sim \mathcal{N}(0, I)$. As $dt \to 0$ these discrete-time dynamics converge to the continuous-time Langevin equation given by $\dot{w}\left(t\right) = -\nabla_w \left(\frac{\gamma}{2}||w(t)||^2 + \mathcal{L}\left(z_w\right)\right) + \sqrt{2T}\xi\left(t\right)$ with $\langle \xi_i(t)\xi_j(t')\rangle = \delta_{ij}\delta\left(t - t'\right)$, so that as $t \to \infty$ the weights will be sampled from the equilibrium distribution in weight space, given by (Risken & Frank, 1996)

$$P\left(w\right) \propto \exp\left(-\frac{1}{T}\left(\frac{\gamma}{2}||w||^2 + \mathcal{L}\left(z_w\right)\right)\right) = \exp\left(-\left(\frac{1}{2\sigma_w^2}||w||^2 + \frac{1}{2\sigma^2}\mathcal{L}\left(z_w\right)\right)\right) \tag{2}$$

The above equality holds since the equilibrium distribution of the Langevin dynamics is also the posterior distribution of a Bayesian neural network (BNN) with an i.i.d. Gaussian prior on the weights $w \sim \mathcal{N}(0, \sigma_w^2 I)$. Thus we can map the hyper-parameters of the training to those of the BNN: $\sigma_w^2 = T/\gamma$ and $\sigma^2 = T/2$. Notice that a sensible scaling for the weight variance at layer $\ell$ is $\sigma_{w,\ell}^2 \sim \mathcal{O}(1/N_{\ell-1})$, thus the weight decay needs to scale as $\gamma_\ell \sim \mathcal{O}(N_{\ell-1})$.

**A transition from weight space to function space** - We aim to move from a distribution over weight space Eq. 2 to a one over function space. Namely, we consider the distribution of $z_w(x)$ implied by the above $P(w)$ where for concreteness we consider a DNN with a single scalar output $z_w(x) \in \mathbb{R}$ on a regression task with data $\{(x_\alpha, y_\alpha)\}_{\alpha=1}^n \subset \mathbb{R}^d \times \mathbb{R}$. Denoting by $P[f]$ the induced measure on function space we formally write

$$P[f] = \int dw \delta[f - z_w]P\left(w\right) \propto e^{-\frac{1}{2\sigma^2}\mathcal{L}[f]} \int dw e^{-\frac{1}{2\sigma_w^2}||w||^2}\delta[f - z_w] \tag{3}$$

where $\int dw$ denotes an integral over all weights and we denote by $\delta[f - z_w]$ a delta-function in function-space. As common in path-integrals or field-theory formalism (Schulman, 2012), such a delta function is understood as a limit procedure where one chooses a suitable basis for function space, trims it to a finite subset, treats $\delta[f - z_w]$ as a product of regular delta-functions, and at the end of the computation takes the size of the subset to infinity.

To proceed we decompose the posterior over functions Eq. 3 as $P[f] \propto e^{-\frac{1}{2\sigma^2}\mathcal{L}[f]}P_0[f]$ where the prior over functions is $P_0[f] \propto \int dw e^{-\frac{1}{2\sigma_w^2}||w||^2}\delta[f - z_w]$. The integration over weights now obtains a clear meaning: it yields the distribution over functions induced by a DNN with i.i.d. random weights chosen according to the prior $P_0(w) \propto e^{-\frac{1}{2\sigma_w^2}||w||^2}$. Thus, we can relate any correlation function in function space and weight space, for instance ($\mathcal{D}f$ is the integration measure over function space)

$$\int \mathcal{D}f P_0[f]f(x)f(x') = \int \mathcal{D}f \int dw P_0(w)\delta[f - z_w]f(x)f(x') = \int dw P_0(w)z_w(x)z_w(x') \tag{4}$$

As noted by Cho & Saul (2009), for highly over-parameterized DNNs the r.h.s. of 4 equals the kernel of the NNGP associated with this DNN, $K(x, x')$. Moreover $P_0[f]$ tends to a Gaussian and can be

written as

$$P_0[f] \propto \exp\left(-\frac{1}{2}\int d\mu(x)d\mu(x')f(x)K^{-1}(x,x')f(x')\right) + \mathcal{O}\left(1/N\right) \tag{5}$$

where $\mu(x)$ is the measure of the input space, and the $\mathcal{O}(1/N)$ scaling of the finite-$N$ correction will be explained in §3. If we now plug 5 in 3, take the loss to be the total square error[1] $\mathcal{L}[f] = \sum_{\alpha=1}^{n}(y_\alpha - f(x_\alpha))^2$, and take $N \to \infty$ we have that the posterior $P[f]$ is that of a GP. Assuming ergodicity, one finds that training-time averaged output of the DNN is given by the posterior mean of a GP, with measurement noise[2] equal to $\sigma^2 = T/2$ and a kernel given by the NNGP of that DNN.

We refer to the above expressions for $P_0[f]$ and $P[f]$ describing the distribution of outputs of a DNN trained according to our protocol – the *NNSP correspondence*. Unlike the NTK correspondence, the kernel which appears here is different and no additional initialization dependent terms appear (as should be the case since we assumed ergodicity). Furthermore, given knowledge of $P_0[f]$ at finite $N$, one can predict the DNN's outputs at finite $N$. Henceforth, we refer to $P_0[f]$ as the prior distribution, as it is the prior distribution of a DNN with random weights drawn from $P_0(w)$.

**Evidence supporting ergodicity** - Our derivation relies on the ergodicity of the dynamics. Ergodicity is in general hard to prove rigorously in non-convex settings, and thus we must revert to heuristics. The most robust evidence of ergodicity in function space is the high level of accuracy of our analytical expressions w.r.t. to our numerical results. This is a self-consistency argument: we assume ergodicity in order to derive our analytical results and then indeed find that they agree very well with the experiment, thus validating our original assumption.

Another indicator of ergodicity is a small auto-correlation time (ACT) of the dynamics. Although short ACT does not logically imply ergodicity (in fact, the converse is true: exponentially long ACT implies non-ergodic dynamics). However, the empirical ACT gives a lower bound on the true correlation time of the dynamics. In our framework, it is sufficient that the dynamics of the outputs $z_w$ be ergodic, even if the dynamics of the weights converge much slower to an equilibrium distribution. Indeed, we have found that the ACTs of the outputs are considerably smaller than those of the weights (see Fig. 2b). Full ergodicity may be too strong of a condition and we don't really need it for our purposes, since we are mainly interested in collecting statistics that will allow us to accurately compute the posterior mean of the distribution in function space. Thus, a weaker condition which is sufficient here is *ergodicity in the mean* (see App. F), and we believe our self-consistent argument above demonstrates that it holds.

In a related manner, optimizing the train loss can be seen as an attempt to find a solution to $n$ constraints using far more variables (roughly $M \cdot N^2$ where $M$ is the number of layers). From a different angle, in a statistical mechanical description of satisfiability problems, one typically expects ergodic behavior when the ratio of the number of variables to the number of constraints becomes much larger than one (Gardner & Derrida, 1988).

## 3 INFERENCE ON THE RESULTING NNSP

Having mapped the time-averaged outputs of a DNN to inference on the above NNSP, we turn to analyze the predictions of this NNSP in the case where $N$ is large but finite, such that the NNSP is only weakly non-Gaussian (i.e. its deviation from a GP is $\mathcal{O}(1/N)$). The main result of this section is a derivation of leading FWCs to the standard GP regression results for the posterior mean $\bar{f}_{\mathrm{GP}}(x_*)$ and variance $\Sigma_{\mathrm{GP}}(x_*)$ on an unseen test point $x_*$, given a training set $\{(x_\alpha, y_\alpha)\}_{\alpha=1}^{n} \subset \mathbb{R}^d \times \mathbb{R}$, namely (Rasmussen & Williams, 2005)

$$\bar{f}_{\mathrm{GP}}(x_*) = \sum_{\alpha,\beta} y_\alpha \tilde{K}_{\alpha\beta}^{-1} K_\beta^*; \qquad \Sigma_{\mathrm{GP}}(x_*) = K^{**} - \sum_{\alpha,\beta} K_\alpha^* \tilde{K}_{\alpha\beta}^{-1} K_\beta^* \tag{6}$$

where $\tilde{K}_{\alpha\beta} := K(x_\alpha, x_\beta) + \sigma^2 \delta_{\alpha\beta}$; $\quad K_\alpha^* := K(x_*, x_\alpha)$; $\quad K^{**} := K(x_*, x_*)$.

---

[1]We take the total error, i.e. we don't divide by $n$ so that $\mathcal{L}[f]$ becomes more dominant for larger $n$.
[2]Here $\sigma^2$ is a property of the training protocol and not of the data itself, or our prior on it.

### 3.1 EDGEWORTH EXPANSION AND PERTURBATION THEORY

Our first task is to find how $P[f]$ changes compared to the Gaussian ($N \to \infty$) scenario. As the data-dependent part $e^{-\mathcal{L}[f]/2\sigma^2}$ is independent of the DNNs, this amounts to obtaining finite width corrections to the prior $P_0[f]$. One way to characterize this is to perform an *Edgeworth expansion* of $P_0[f]$ (Mccullagh, 2017; Sellentin et al., 2017). We give a short recap of the Edgeworth expansion to elucidate our derivation, beginning with a scalar valued RV. Consider continuous iid RVs $\{Z_i\}$ and assume WLOG $\langle Z_i \rangle = 0$, $\langle Z_i^2 \rangle = 1$, with higher cumulants $\kappa_r^Z$ for $r \geq 3$. Now consider their normalized sum $Y_N = \frac{1}{\sqrt{N}} \sum_{i=1}^{N} Z_i$. From additivity and homogeneity of cumulants we have $\kappa_{r \geq 2} := \kappa_{r \geq 2}^Y = \frac{N \kappa_r^Z}{(\sqrt{N})^r} = \frac{\kappa_r^Z}{N^{r/2-1}}$. Now, let $\varphi(y) := (2\pi)^{-1/2} e^{-y^2/2}$. The *characteristic function* of $Y$ is $\hat{f}(t) := \mathcal{F}[f(y)] = \exp\left(\sum_{r=1}^{\infty} \kappa_r \frac{(it)^r}{r!}\right) = \exp\left(\sum_{r=3}^{\infty} \kappa_r \frac{(it)^r}{r!}\right) \hat{\varphi}(t)$. Taking the inverse Fourier transform $\mathcal{F}^{-1}$ has the effect of mapping $it \mapsto -\partial_y$ thus we get $f(y) = \exp\left(\sum_{r=3}^{\infty} \kappa_r \frac{(-\partial_y)^r}{r!}\right) \varphi(y) = \varphi(y)\left(1 + \sum_{r=3}^{\infty} \frac{\kappa_r}{r!} H_r(y)\right)$ where $H_r(y)$ is the $r$th *Hermite polynomial*, e.g. $H_4(y) = y^4 - 6y^2 + 3$. If we were to consider vector-valued RVs, then the $r$'th order cumulant becomes a tensor with $r$ indices, and the Hermite polynomials become multi-variate polynomials. In our case, we are considering *random functions* defined by our stochastic process (the NNSP), thus the cumulants are functional tensors, i.e. are continuously indexed by the inputs $x_\alpha$.

This is especially convenient here since for all DNNs with the last layer being fully-connected, all odd cumulants vanish and the $2r^{\text{th}}$ cumulant scales as $1/N^{r-1}$. Consequently, at large $N$ we can characterize $P_0[f]$ up to $\mathcal{O}(N^{-2})$ by its second and fourth cumulants, $K(x_1, x_2)$ and $U(x_1, x_2, x_3, x_4)$, respectively. Thus the leading order correction to $P_0[f]$ reads

$$P_0[f] \propto e^{-S_{\text{GP}}[f]}\left(1 - \frac{1}{N} S_U[f]\right) + \mathcal{O}\left(1/N^2\right) \tag{7}$$

where the GP action $S_{\text{GP}}$ and the first FWC action $S_U$ are given by

$$S_{\text{GP}}[f] = \frac{1}{2} \int d\mu_{1:2} f_{x_1} K_{x_1,x_2}^{-1} f_{x_2}; \qquad S_U[f] = -\frac{1}{4!} \int d\mu_{1:4} U_{x_1,x_2,x_3,x_4} H_{x_1,x_2,x_3,x_4}[f] \tag{8}$$

Here, $H$ is the 4th functional Hermite polynomial (see App. A), $U$ is the 4th order functional cumulant of the NN output[3], which depends on the choice of the activation function $\phi$

$$U_{x_1,x_2,x_3,x_4} = \varsigma_a^4\left(\langle \phi_\alpha \phi_\beta \phi_\gamma \phi_\delta \rangle - \langle \phi_\alpha \phi_\beta \rangle \langle \phi_\gamma \phi_\delta \rangle\right) + 2 \text{ other perms. of } (\alpha, \beta, \gamma, \delta) \in \{1, \ldots, 4\} \tag{9}$$

where $\phi_\alpha := \phi(z_i^{\ell-1}(x_\alpha))$ and the pre-activations are $z_i^\ell(x) = b_i^\ell + \sum_{j=1}^{N_\ell} W_{ij} \phi(z_j^{\ell-1}(x))$. Here we distinguished between the scaled and non-scaled weight variances: $\sigma_a^2 = \varsigma_a^2/N$, where $a$ are the weights of the last layer. Our shorthand notation for the integration measure over inputs means e.g. $d\mu_{1:4} := d\mu(x_1) \cdots d\mu(x_4)$.

Using perturbation theory, in App. B we compute the leading FWC to the posterior mean $\bar{f}(x_*)$ and variance $\langle (\delta f(x_*))^2 \rangle$ on a test point $x_*$

$$\bar{f}(x_*) = \bar{f}_{\text{GP}}(x_*) + N^{-1} \bar{f}_U(x_*) + \mathcal{O}(N^{-2})$$
$$\langle (\delta f(x_*))^2 \rangle = \Sigma_{\text{GP}}(x_*) + N^{-1} \Sigma_U(x_*) + \mathcal{O}(N^{-2}) \tag{10}$$

with $\Sigma_U(x_*) = \langle (f(x_*))^2 \rangle_U - 2\bar{f}_{\text{GP}}(x_*) \bar{f}_U(x_*)$ and

$$\bar{f}_U(x_*) = \frac{1}{6} \tilde{U}_{\alpha_1 \alpha_2 \alpha_3}^* \left(\tilde{y}_{\alpha_1} \tilde{y}_{\alpha_2} \tilde{y}_{\alpha_3} - 3\tilde{K}_{\alpha_1 \alpha_2}^{-1} \tilde{y}_{\alpha_3}\right)$$
$$\langle (f(x_*))^2 \rangle_U = \frac{1}{2} \tilde{U}_{\alpha_1 \alpha_2}^{**} \left(\tilde{y}_{\alpha_1} \tilde{y}_{\alpha_2} - \tilde{K}_{\alpha_1 \alpha_2}^{-1}\right) \tag{11}$$

where all repeating indices are summed over the training set (i.e. range over $\{1, \ldots, n\}$), denoting: $\tilde{y}_\alpha := \tilde{K}_{\alpha\beta}^{-1} y_\beta$, and defining

$$\tilde{U}_{\alpha_1 \alpha_2 \alpha_3}^* := U_{\alpha_1 \alpha_2 \alpha_3}^* - U_{\alpha_1 \alpha_2 \alpha_3 \alpha_4} \tilde{K}_{\alpha_4 \beta}^{-1} K_\beta^*$$
$$\tilde{U}_{\alpha_1 \alpha_2}^{**} := U_{\alpha_1 \alpha_2}^{**} - \left(U_{\alpha_1 \alpha_2 \alpha_3}^* + \tilde{U}_{\alpha_1 \alpha_2 \alpha_3}^*\right) \tilde{K}_{\alpha_3 \beta}^{-1} K_\beta^* \tag{12}$$

---

[3] Here we take $U \sim \mathcal{O}(1)$ to emphasize the scaling with $N$ in Eqs. 7, 10.

Equations 11, 12 are one of our key analytical results, which are qualitatively different from the corresponding GP expressions Eq. 6. The correction to the predictive mean $\bar{f}_U(x_*)$ has a linear term in $y$, which can be viewed as a correction to the GP kernel, but also a cubic term in $y$, unlike $\bar{f}_{\mathrm{GP}}(x_*)$ which is purely linear. The correction to the predictive variance $\Sigma_U(x_*)$ has quartic and quadratic terms in $y$, unlike $\Sigma_{\mathrm{GP}}(x_*)$ which is $y$-independent. $\tilde{U}^*_{\alpha_1\alpha_2\alpha_3}$ has a clear interpretation in terms of GP regression: if we consider the indices $\alpha_1, \alpha_2, \alpha_3$ as fixed, then $U^*_{\alpha_1\alpha_2\alpha_3}$ can be thought of as the ground truth value of a target function (analogous to $y_*$), and the second term on the r.h.s. $U_{\alpha_1\alpha_2\alpha_3\alpha_4}\tilde{K}^{-1}_{\alpha_4\beta}K^*_\beta$ is then the GP prediction of $U^*_{\alpha_1\alpha_2\alpha_3}$ with the kernel $K$, where $\alpha_4$ runs on the training set (compare to $\bar{f}_{\mathrm{GP}}(x_*)$ in Eq. 6). Thus $\tilde{U}^*_{\alpha_1\alpha_2\alpha_3}$ is the *discrepancy* in predicting $U_{\alpha_1\alpha_2\alpha_3\alpha_4}$ using a GP with kernel $K$. In §3.2 we study the behavior of $\bar{f}_U(x_*)$ as a function of $n$.

The posterior variance $\Sigma(x) = \left\langle (\delta f(x))^2 \right\rangle$ has a clear interpretation in our correspondence: it is a measure of how much we can decrease the test loss by averaging. Our procedure for generating empirical network outputs involves time-averaging over the training dynamics after reaching equilibrium and also over different realizations of noise and initial conditions (see App. F). This allows for a reliable comparison with our FWC theory for the mean. In principle, one could use the network outputs at the end of training without this averaging, in which case there will be fluctuations that will scale with $\Sigma(x_\alpha)$. Following this, one finds that the expected MSE test loss after training saturates is $n_*^{-1}\sum_{\alpha=1}^{n_*}\left(\left\langle \left(\bar{f}(x_\alpha) - y(x_\alpha)\right)^2 \right\rangle + \Sigma(x_\alpha)\right)$ where $n_*$ is the size of the test set.

## 3.2 FINITE WIDTH CORRECTIONS FOR SMALL AND LARGE DATA SETS

The expressions in Eqs. 6, 11 for the GP prediction and the leading FWC are explicit but only up to a potentially large matrix inversion, $\tilde{K}^{-1}$. These matrices also have a random component related to the largely arbitrary choice of the particular $n$ training points used to characterize the target function. An insightful tool, used in the context of GPs, which solves both these issues is the *Equivalent Kernel* (EK) (Rasmussen & Williams, 2005; Sollich & Williams, 2004). The EK approximates the GP predictions at large $n$, after averaging on all draws of (roughly) $n$ training points representing the target function. Even if one is interested in a particular dataset, the EK result captures the behavior of specific dataset up to small corrections. Essentially, the discrete sums over the training set appearing in Eq. 6 are replaced by integrals over all input space, which together with a spectral decomposition of the kernel function $K(x, x') = \sum_i \lambda_i \psi_i(x)\psi_i(x')$ yields the well known result

$$\bar{f}^{\mathrm{EK}}_{\mathrm{GP}}(x_*) = \int d\mu(x') \sum_i \frac{\lambda_i \psi_i(x_*)\psi_i(x')}{\lambda_i + \sigma^2/n} g(x') \tag{13}$$

Here we develop an extension of Eq. 13 for the NNSPs we find at large but finite $N$. In particular, we find the leading non-linear correction to the EK result, i.e. the "EK analogue" of Eq. 11. To this end, we consider the average predictions of an NNSP trained on an ensemble of data sets of size $n'$, corresponding to $n'$ independent draws from a distribution $\mu(x)$ over all possible inputs $x$. Following the steps in App. J we find

$$\bar{f}^{\mathrm{EK}}_U(x_*) = \frac{1}{6}\tilde{\delta}_{x_*x_1}U_{x_1,x_2,x_3,x_4}\left\{\frac{n^3}{\sigma^6}\tilde{\delta}_{x_2x'_2}g(x'_2)\tilde{\delta}_{x_3x'_3}g(x'_3)\tilde{\delta}_{x_4x'_4}g(x'_4) - \frac{3n^2}{\sigma^4}\tilde{\delta}_{x_2,x_3}\tilde{\delta}_{x_4,x'_4}g(x'_4)\right\} \tag{14}$$

where an integral $\int d\mu(x)$ is implicit for every pair of repeated $x$ coordinates. We introduced the *discrepancy operator* $\tilde{\delta}_{xx'}$ which acts on some function $\varphi$ as $\int d\mu(x')\tilde{\delta}_{xx'}\varphi(x') := \tilde{\delta}_{xx'}\varphi(x') = \varphi(x) - \bar{f}^{\mathrm{EK}}_{\mathrm{GP}}(x)$. Essentially, Eq. 14 is derived from Eq. 11 by replacing each $\tilde{K}^{-1}$ by $(n/\sigma^2)\tilde{\delta}$ and noticing that in this regime $\tilde{U}^*_{x_2,x_3,x_4}$ in Eq. 12 becomes $\tilde{\delta}_{x_*x_1}U_{x_1,x_2,x_3,x_4}$. Interestingly, $\bar{f}^{\mathrm{EK}}_U(x_*)$ is written explicitly in terms of meaningful quantities: $\tilde{\delta}_{xx'}g(x')$ and $\tilde{\delta}_{x_*x_1}U_{x_1,x_2,x_3,x_4}$.

Equations 13, 14 are valid for any weakly non-Gaussian process, including ones related to CNNs (where $N$ corresponds to the number of channels). It can also be systematically extended to smaller values of $n$ by taking into account higher terms in $1/n$, as in Cohen et al. (2019). At $N \to \infty$, we obtain the standard EK result, Eq. 13. It is basically a high-pass linear filter which filters out features of $g$ that have support on eigenfunctions $\psi_i$ associated with eigenvalues $\lambda_i$ that are small relative to $\sigma^2/n$. We stress that the $\psi_i, \lambda_i$'s are independent of any particular size $n$ dataset but rather are

a property of the average dataset. In particular, no computationally costly data dependent matrix inversion is needed to evaluate Eq. 13.

Turning to our FWC result, Eq. 14, it depends on $g(x)$ only via the discrepancy operator $\tilde{\delta}_{xx'}$. Thus these FWCs would be proportional to the error of the DNN, at $N \to \infty$. In particular, perfect performance at $N \to \infty$, implies no FWC. Second, the DNN's average predictions act as a linear transformation on the target function combined with a cubic non-linearity. Third, for $g(x)$ having support only on some finite set of eigenfunctions $\psi_i$ of $K$, $\tilde{\delta}_{xx'}g(x')$ would scale as $\sigma^2/n$ at very large $n$. Thus the above cubic term would lose its explicit dependence on $n$. The scaling with $n$ of this second term is less obvious, but numerical results suggest that $\tilde{\delta}_{x_2 x_3}$ also scales as $\sigma^2/n$, so that the whole expression in the $\{\cdots\}$ has no scaling with $n$. In addition, some decreasing behavior with $n$ is expected due to the $\tilde{\delta}_{x_* x_1} U_{x_1, x_2, x_3, x_4}$ factor which can be viewed as the discrepancy in predicting $U_{x, x_2, x_3, x_4}$, at fixed $x_2, x_3, x_4$, based on $n$ random samples ($x_\alpha$'s) of $U_{x_\alpha, x_2, x_3, x_4}$. In Fig. 1 we illustrate this behavior at large $n$ and also find that for small $n$ the FWC is small but increasing with $n$, implying that at large $N$ FWCs are only important at intermediate values of $n$.

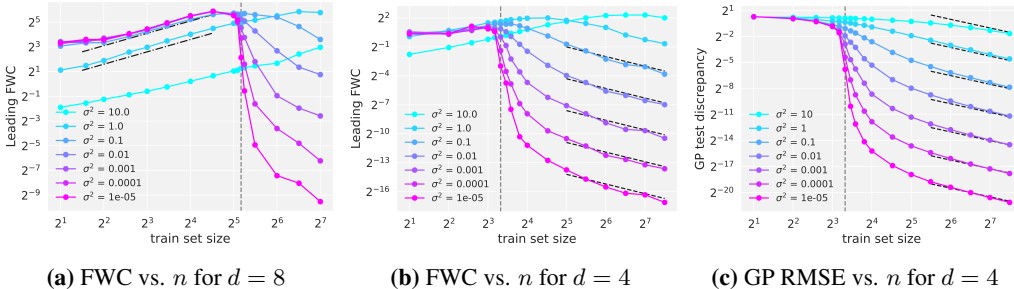

**(a)** FWC vs. $n$ for $d = 8$    **(b)** FWC vs. $n$ for $d = 4$    **(c)** GP RMSE vs. $n$ for $d = 4$

**Figure 1: Leading FWC to the mean $|\bar{f}_U(x_*)|$ (Eq. 11) and GP discrepancy (RMSE) as a function of train set size $n$ for varying training noise $\sigma^2$.** The target is quadratic $g(x) = x^\mathsf{T} A x = \mathcal{O}(1)$ with $x \in \mathbb{S}_{d-1}(\sqrt{d})$ so the number of parameters to be learnt is $d(d+1)/2$ (vertical grey dashed line). The GP discrepancy is monotonically decreasing with $n$ whereas $|\bar{f}_U(x_*)|$ increases linearly for small $n$ (dashed-dotted lines in (a)) before it decays (best illustrated for larger $d$ and $\sigma^2$). For sufficiently large $n$, both the GP discrepancy and $|\bar{f}_U(x_*)|$ scale as $1/n$ (diagonal dashed black lines in (b),(c)). This verifies our prediction for the scaling of FWCs with $n$, Eq. 14 in the large $n$ regime . Notably, it implies that at large $N$ FWCs are only important at intermediate values of $n$.

## 4 NUMERICAL EXPERIMENTS

In this section we numerically test our analytical results. We first demonstrate that in the limit $N \to \infty$ the outputs of a FCN trained in the regime of the NNSP correspondence converge to a GP with a known kernel, and that the MSE between them scales as $\sim 1/N^2$ which is the scaling of the leading FWC squared. Second, we show that introducing the leading FWC term $N^{-1}\bar{f}_U(x_*)$, Eq. 11, further reduces this MSE by more than an order of magnitude. Third, we study the performance gap between finite CNNs and their corresponding NNGPs on CIFAR-10.

### 4.1 TOY EXAMPLE: FULLY CONNECTED NETWORKS ON SYNTHETIC DATA

We trained a 2-layer FCN $f(x) = \sum_{i=1}^{N} a_i \phi(w^{(i)} \cdot x)$ on a quadratic target $y(x) = x^\mathsf{T} A x$ where the $x$'s are sampled with a uniform measure from the hyper-sphere $\mathbb{S}_{d-1}(\sqrt{d})$, see App. G.1 for more details. Our settings are such that there are not enough training points to fully learn the target: Fig. 2a shows that the time averaged outputs (after reaching equilibrium) $\bar{f}_{\text{DNN}}(x_*)$ is much closer to the GP prediction $\bar{f}_{\text{GP}}(x_*)$ than to the ground truth $y_*$. Otherwise, the convergence of the network output to the corresponding NNGP as $N$ grows (shown in Fig. 2c) would be trivial, since all reasonable estimators would be close to the target and hence close to each other.

In Fig. 2c we plot in log-log scale (with base 10) the MSE (normalized by $(\bar{f}_{\text{DNN}})^2$) between the predictions of the network $\bar{f}_{\text{DNN}}$ and the corresponding GP and FWC predictions for quadratic and ReLU activations. We find that indeed for sufficiently large widths ($N \gtrsim 500$) the slope of the GP-DNN MSE approaches $-2.0$ (for both ReLU and quadratic), which is expected from our theory, since the leading FWC scales as $1/N$. For smaller widths, higher order terms (in $1/N$) in the Edgeworth series Eq. 7 come into play. For quadratic activation, we find that our FWC result further reduces the MSE by more than an order of magnitude relative to the GP theory. We recognize a regime where the GP and FWC MSEs intersect at $N \lesssim 100$, below which our FWC actually increases the MSE, which suggests a scale of how large $N$ needs to be for our leading FWC theory to hold.

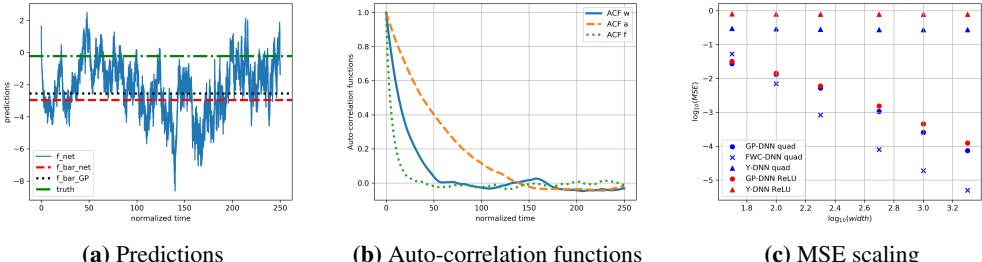

**(a)** Predictions      **(b)** Auto-correlation functions      **(c)** MSE scaling

**Figure 2: Fully connected 2-layer network trained on a regression task. (a)** Network outputs on a test point $f(x_*, t)$ vs. normalized time: the time-averaged DNN output $\bar{f}(x_*)$ (dashed line) is much closer to the GP prediction $\bar{f}_{\text{GP}}(x_*)$ (dotted line) than to the ground truth $y_*$ (dashed-dotted line). **(b)** ACFs of the time series of the 1st and 2nd layer weights, and of the outputs: the output converges to equilibrium faster than the weights. **(c)** Relative MSE between the network outputs and the labels $y$ (triangles), GP predictions $\bar{f}_{\text{GP}}(x_*)$ Eq. 6 (dots), and FWC predictions Eq. 10 (x's), shown vs. width for quadratic (blue) and ReLU (red) activations. For sufficiently large widths ($N \gtrsim 500$) the slope of the GP-DNN MSE approaches $-2.0$ and the FWC-DNN MSE is further improved by more than an order of magnitude.

## 4.2 PERFORMANCE GAP BETWEEN FINITE CNNs AND THEIR NNGP

Several papers have shown that the performance on image classification tasks of SGD-trained finite CNNs can surpass that of the corresponding GPs, be it NTK (Arora et al., 2019) or NNGP (Novak et al., 2018). More recently, Lee et al. (2020) emphasized that this performance gap depends on the procedure used to collapse the spatial dimensions of image-shaped data before the final readout layer: flattening the image into a one-dimensional vector (CNN-VEC) or applying global average pooling to the spatial dimensions (CNN-GAP). It was observed that while infinite FCN and CNN-VEC outperform their respective finite networks, infinite CNN-GAP networks under-perform their finite-width counterparts, i.e. there exists a finite optimal width.

One notable margin, of about 15% accuracy on CIFAR10, was shown in Novak et al. (2018) for the case of CNN-GAP. It was further pointed out there, that the NNGPs associated with CNN-VEC, coincide with those of the corresponding Locally Connected Networks (LCNs), namely CNNs without weight sharing between spatial locations. Furthermore, the performance of SGD-trained LCNs was found to be on par with that of their NNGPs. We argue that our framework can account for this observation. The priors $P_0[f]$ of a LCN and CNN-VEC agree on their second cumulant (the covariance), which is the only one not vanishing as $N \to \infty$, but they need not agree on their higher order cumulants, which come into play at finite $N$. In App. I we show that $U$ appearing in our leading FWC, already differentiates between CNNs and LCNs. Common practice strongly suggests that the prior over functions induced by CNNs is better suited than that of LCNs for classification of natural images. As a result we expect that the test loss of a finite-width CNN trained using our protocol will initially decrease with $N$ but then increase beyond some *optimal width* $N_{\text{opt}}$, tending towards the loss of the corresponding GP as $N \to \infty$. This is in contrast to SGD behavior reported in some works where the CNN performance seems to saturate as a function of $N$, to some value better than

the NNGP (Novak et al., 2018; Neyshabur et al., 2018). Notably those works used maximum over architecture scans, high learning rates, and early stopping, all of which are absent from our training protocol.

To test the above conjecture we trained, according to our protocol, a CNN with six convolutional layers and two fully connected layers on CIFAR10, and used CNN-VEC for the readout. We used MSE loss with a one-hot encoding into a 10 dimensional vector of the categorical label; further details and additional settings are given in App. G. Fig. 3 demonstrates that, using our training protocol, a finite CNN can outperform its corresponding GP and approaches its GP as the number of channels increases. This phenomenon was observed in previous studies under realistic training settings (Novak et al., 2018), and here we show that it appears also under our training protocol. We note that a similar yet more pronounced trend in performance appears here also when one considers the averaged MSE loss rather the the MSE loss of the average outputs.

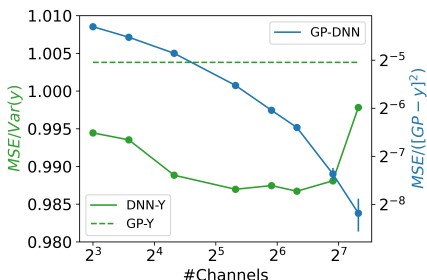

**Figure 3:** DNN-GP MSE demonstrates convergence to a slope of $-2.0$, validating the theoretically expected scaling. DNN-ground truth (Y) MSE shows finite CNN can outperform corresponding GP.

## 5 CONCLUSION

In this work we presented a correspondence between finite-width DNNs trained using Langevin dynamics (i.e. using small learning rates, weight-decay and noisy gradients) and inference on a stochastic-process (the NNSP), which approaches the NNGP as $N \to \infty$. We derived finite width corrections, that improve upon the accuracy of the NNGP approximation for predicting the DNN outputs on unseen test points, as well as the expected fluctuations around these. In the limit of a large number of training points $n \to \infty$, explicit expressions for the DNNs' outputs were given, involving no costly matrix inversions. In this regime, the FWC can be written in terms of the discrepancy of GP predictions, so that when GP has a small test error the FWC will be small, and vice versa. In the small $n$ regime, the FWC is small but grows with $n$, which implies that at large $N$, FWCs are only important at intermediate values of $n$. For no-pooling CNNs, we build on an observation made by Novak et al. (2018) that finite CNNs outperform their corresponding NNGPs, and show that this is because the leading FWCs reflect the weight-sharing property of CNNs which is ignored at the level of the NNGP. This constitutes one real-world example where the FWC is well suited to the structure of the data distribution, and thus improves performance relative to the corresponding GP. In a future study, it would be very interesting to consider well controlled toy models that can elucidate under what conditions on the architecture and data distribution does the FWC improve performance relative to GP.

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

## A    EDGEWORTH SERIES

The Central Limit Theorem (CLT) tells us that the distribution of a sum of $N$ independent RVs will tend to a Gaussian as $N \to \infty$. Its relevancy for wide fully-connected DNNs (or CNNs with many channels) comes from the fact that every pre-activation averages over $N$ uncorrelated random variables thereby generating a Gaussian distribution at large $N$ (Cho & Saul, 2009), augmented by higher order cumulants which decay as $1/N^{r/2-1}$, where $r$ is the order of the cumulant. When higher order cumulants are small, an Edgeworth series (see e.g. Mccullagh (2017); Sellentin et al. (2017)) is a useful practical tool for obtaining the probability distribution from these cumulants. Having the probability distribution and interpreting its logarithm as our action, places us closer to standard field-theory formalism.

For simplicity we focus on a 2-layer network, but the derivation generalizes straightforwardly to networks of any depth. We are interested in the finite $N$ corrections to the prior distribution $P_0[f]$, i.e. the distribution of the DNN output $f(x) = \sum_{i=1}^{N} a_i \phi(w_i^\mathsf{T} x)$, with $a_i \sim \mathcal{N}(0, \frac{\varsigma_a^2}{N})$ and $w_i \sim \mathcal{N}(\mathbf{0}, \frac{\varsigma_w^2}{d} I)$. Because $a$ has zero mean and a variance that scales as $1/N$, all odd cumulants are zero and the $2r$'th cumulant scales as $1/N^{r-1}$. This holds true for any DNN having a fully-connected last layer with variance scaling as $1/N$. The derivation of the multivariate Edgeworth series can be found in e.g. Mccullagh (2017); Sellentin et al. (2017), and our case is similar where instead of a vector-valued RV we have the functional RV $f(x)$, so the cumulants become "functional tensors" i.e. multivariate functions of the input $x$. Thus, the leading FWC to the prior $P_0[f]$ is

$$P_0[f] = \frac{1}{Z} e^{-S_{\mathrm{GP}}[f]} \left[ 1 + \frac{1}{4!} \int d\mu(x_1) \cdots d\mu(x_4) \, U(x_1, x_2, x_3, x_4) \, H[f; \, x_1, x_2, x_3, x_4] \right] + \mathcal{O}(1/N^2)$$
(A.1)

where $S_{\mathrm{GP}}[f]$ is as in the main text Eq. 8 and the 4th Hermite functional tensor is

$$H[f] = \int d\mu(x_1') \cdots d\mu(x_4') \, K^{-1}(x_1, x_1') \cdots K^{-1}(x_4, x_4') \, f(x_1') \cdots f(x_4')$$

$$- K^{-1}(x_\alpha, x_\beta) \int d\mu(x_\mu') \, d\mu(x_\nu') \, K^{-1}(x_\mu, x_\mu') \, K^{-1}(x_\nu, x_\nu') \, f(x_\mu') \, f(x_\nu') \, [6] \quad (\text{A.2})$$

$$+ K^{-1}(x_\alpha, x_\beta) \, K^{-1}(x_\mu, x_\nu) \, [3]$$

where by the integers in $[\cdot]$ we mean all possible combinations of this form, e.g.

$$K_{\alpha\beta}^{-1} K_{\mu\nu}^{-1} = K_{12}^{-1} K_{34}^{-1} + K_{13}^{-1} K_{24}^{-1} + K_{14}^{-1} K_{23}^{-1}$$
(A.3)

$H[f]$ is the functional analogue of the fourth Hermite polynomial: $H_4(x) = x^4 - 6x^2 + 3$, which appears in the scalar Edgeworth series expanded about a standard Gaussian.

## B    FIRST ORDER CORRECTION TO POSTERIOR MEAN AND VARIANCE

### B.1    POSTERIOR MEAN

The posterior mean with the leading FWC action is given by

$$\langle f(x_*) \rangle = \frac{\int \mathcal{D}f e^{-S[f]} f(x_*)}{\int \mathcal{D}f e^{-S[f]}} + \mathcal{O}(1/N^2)$$
(B.1)

where

$$S[f] = S_{\mathrm{GP}}[f] + S_{\mathrm{Data}}[f] + S_U[f]; \qquad S_{\mathrm{Data}}[f] = \frac{1}{2\sigma^2} \sum_{\alpha=1}^{n} (f(x_\alpha) - y_\alpha)^2 \qquad (\text{B.2})$$

where the $\mathcal{O}(1/N^2)$ implies that we only treat the first order Taylor expansion of $S[f]$, and where $S_{\mathrm{GP}}[f], S_U[f]$ are as in the main text Eq. 8. The general strategy is to bring the path integral $\int \mathcal{D}f$ to the front, so that we will get just correlation functions w.r.t. the Gaussian theory (including the data term $S_{\mathrm{Data}}[f]$) $\langle \cdots \rangle_0$, namely the well known results (Rasmussen & Williams, 2005) for

$\bar{f}_{\mathrm{GP}}(x_*) = \langle f(x_*)\rangle_0$ and $\Sigma_{\mathrm{GP}}(x_*) = \langle(\delta f(x_*))^2\rangle_0$, and then finally perform the integrals over input space. Expanding both the numerator and the denominator of Eq. B.1, the leading finite width correction for the posterior mean reads

$$\bar{f}_U(x_*) = \frac{1}{4!}\left(\int d\mu_{1:4}U(x_1, x_2, x_3, x_4)\langle f(x_*)H[f]\rangle_0 - \langle f(x_*)\rangle_0\int d\mu_{1:4}U(x_1, x_2, x_3, x_4)\langle H[f]\rangle_0\right)$$
(B.3)

This, as standard in field theory, amounts to omitting all terms corresponding to bubble diagrams, namely we keep only terms with a factor of $\langle f(x_*)f(x'_\alpha)\rangle_0$ and ignore terms with a factor of $\langle f(x_*)\rangle_0$, since these will cancel out. This is a standard result in perturbative field theory (see e.g. Zee (2003)).

We now write down the contributions of the quartic, quadratic and constant terms in $H[f]$:

1. For the quartic term in $H[f]$, we have

$$\langle f(x_*)f(x'_1)f(x'_2)f(x'_3)f(x'_4)\rangle_0 - \langle f(x_*)\rangle_0\langle f(x'_1)f(x'_2)f(x'_3)f(x'_4)\rangle_0$$
$$= \Sigma(x_*, x'_\alpha)\Sigma(x'_\beta, x'_\gamma)\bar{f}(x'_\delta)[12] + \Sigma(x_*, x'_\alpha)\bar{f}(x'_\beta)\bar{f}(x'_\gamma)\bar{f}(x'_\delta)[4]$$
(B.4)

   We dub these terms by $\bar{f}\Sigma\Sigma_*$ and $\bar{f}\bar{f}\bar{f}\Sigma_*$ to be referenced shortly. We mention here that they are the source of the linear and cubic terms in the target $y$ appearing in Eq. 11 in the main text.

2. For the quadratic term in $H[f]$, we have

$$\langle f(x_*)f(x'_\mu)f(x'_\nu)\rangle_0 - \langle f(x_*)\rangle_0\langle f(x'_\mu)f(x'_\nu)\rangle_0 = \Sigma(x_*, x'_\mu)\bar{f}(x'_\nu)[2]$$
(B.5)

   we note in passing that these cancel out exactly together with similar but opposite sign terms/diagrams in the quartic contribution, which is a reflection of measure invariance. This is elaborated on in §B.3.

3. For the constant terms in $H[f]$, we will be left only with bubble diagram terms $\propto \int \mathcal{D}f\, f(x_*)$ which will cancel out in the leading order of $1/N$.

## B.2 Posterior variance

The posterior variance is given by

$$\Sigma(x_*) = \langle f(x_*)f(x_*)\rangle - \bar{f}^2$$
$$= \langle f(x_*)f(x_*)\rangle_0 + \langle f(x_*)f(x_*)\rangle_U - \bar{f}_{\mathrm{GP}}^2 - 2\bar{f}_{\mathrm{GP}}\bar{f}_U + \mathcal{O}(1/N^2)$$
(B.6)
$$= \Sigma_{\mathrm{GP}}(x_*) + \langle f(x_*)f(x_*)\rangle_U - 2\bar{f}_{\mathrm{GP}}\bar{f}_U + \mathcal{O}(1/N^2)$$

Following similar steps as for the posterior mean, the leading finite width correction for the posterior second moment at $x_*$ reads

$$\langle f(x_*)f(x_*)\rangle_U =$$
$$\frac{1}{4!}\left(\int d\mu_{1:4}U(x_1, x_2, x_3, x_4)\langle f(x_*)f(x_*)H[f]\rangle_0 - \langle f(x_*)f(x_*)\rangle_0\int d\mu_{1:4}U(x_1, x_2, x_3, x_4)\langle H[f]\rangle_0\right)$$
(B.7)

As for the posterior mean, the constant terms in $H[f]$ cancel out and the contributions of the quartic and quadratic terms are

$$\text{quartic terms} = \Sigma_{*\alpha}\Sigma_{*\beta}\bar{f}_\gamma\bar{f}_\delta[12] + \Sigma_{*\alpha}\Sigma_{*\beta}\Sigma_{\gamma\delta}[12]$$
(B.8)

and

$$\text{quadratic terms} = \Sigma_{*\mu}\Sigma_{*\nu}[2]$$
(B.9)

### B.3 MEASURE INVARIANCE OF THE RESULT

The expressions derived above may seem formidable, since they contain many terms and involve integrals over input space which seemingly depend on the measure $\mu(x)$. Here we show how they may in fact be simplified to the compact expressions in the main text Eq. 11 which involve only discrete sums over the training set and no integrals, and are thus manifestly measure-invariant.

For simplicity, we show here the derivation for the FWC of the mean $\bar{f}_U(x_*)$, and a similar derivation can be done for $\Sigma_U(x_*)$. In the following, we carry out the $x$ integrals, by plugging in the expressions from Eq. 6 and coupling them to $U$. As in the main text, we use the Einstein summation notation, i.e. repeated indices are summed over the training set. The contribution of the quadratic terms is

$$A_{\alpha_1,*}\tilde{K}^{-1}_{\alpha_1\beta_1}y_{\beta_1} - A_{\alpha_1\alpha_2}\tilde{K}^{-1}_{\alpha_1\beta_1}\tilde{K}^{-1}_{\alpha_2\beta_2}y_{\beta_1}K_{\beta_2,*} \tag{B.10}$$

where we defined

$$A(x_3, x_4) := \iint d\mu(x_1)d\mu(x_2)U(x_1, x_2, x_3, x_4)K^{-1}(x_1, x_2) \tag{B.11}$$

Fortunately, this seemingly measure-dependent expression will cancel out with one of the terms coming from the $\bar{f}\Sigma\Sigma_*$ contribution of the quartic terms in $H[f]$. This is not a coincidence and is a general feature of the Hermite polynomials appearing in the Edgeworth series, thus for any order in $1/N$ in the Edgeworth series we will always be left only with measure invariant terms. Collecting all terms that survive we have

$$\frac{1}{4!}\left\{4\tilde{U}^*_{\alpha_1\alpha_2\alpha_3}\tilde{K}^{-1}_{\alpha_1\beta_1}\tilde{K}^{-1}_{\alpha_2\beta_2}\tilde{K}^{-1}_{\alpha_3\beta_3}y_{\beta_1}y_{\beta_2}y_{\beta_3} - 12\tilde{U}^*_{\alpha_1\alpha_2\alpha_3}\tilde{K}^{-1}_{\alpha_2\beta_2}\tilde{K}^{-1}_{\alpha_1\beta_1}y_{\beta_1}\right\} \tag{B.12}$$

where we defined

$$\tilde{U}^*_{\alpha_1\alpha_2\alpha_3} := U^*_{\alpha_1\alpha_2\alpha_3} - U_{\alpha_1\alpha_2\alpha_3\alpha_4}\tilde{K}^{-1}_{\alpha_4\beta_4}K^*_{\beta_4} \tag{B.13}$$

This is a more explicit form of the result reported in the main text, Eq. 11.

## C FINITE WIDTH CORRECTIONS FOR MORE THAN ONE HIDDEN LAYER

For simplicity, consider a fully connected network with two hidden layers both of width $N$, and no biases, thus the pre-activations $h(x)$ and output $z(x)$ are given by

$$\begin{aligned} h(x) &= \frac{\sigma_{w_2}}{\sqrt{N}}W^{(2)}\phi\left(\frac{\sigma_{w_1}}{\sqrt{d}}W^{(1)}x\right) \\ z(x) &= \frac{\sigma_a}{\sqrt{N}}a^{\mathsf{T}}\phi\left(h^{(2)}(x)\right) \end{aligned} \tag{C.1}$$

We want to find the 2nd and 4th cumulants of $z(x)$. Recall that we found that the leading order Edgeworth expansion for the functional distribution of $h$ is

$$P_{K,U}[h] \propto e^{-\frac{1}{2}h(x_1')K^{-1}(x_1',x_2')h(x_2')}\left(1 + \frac{1}{N}U(x_1',x_2',x_3',x_4')H[h;x_1',x_2',x_3',x_4']\right) \tag{C.2}$$

where $K^{-1}(x_1', x_2')$ and $U(x_1', x_2', x_3', x_4')$ are known from the previous layer. So we are looking for two maps:

$$\begin{aligned} \mathcal{K}_\phi(K, U)(x, x') &= \langle\phi(h(x))\phi(h(x'))\rangle_{P_{K,U}[h]} \\ \mathcal{U}_\phi(K, U)(x_1, x_2, x_3, x_4) &= \langle\phi(h(x_1))\phi(h(x_2))\phi(h(x_3))\phi(h(x_4))\rangle_{P_{K,U}[h]} \end{aligned} \tag{C.3}$$

so that the mapping between the first two cumulants $K$ and $U$ of two consequent layers is (assuming no biases)

$$\begin{aligned} \frac{K^{(\ell+1)}(x, x')}{\sigma^2_{w^{(\ell+1)}}} &= \mathcal{K}_\phi\left(K^{(\ell)}, U^{(\ell)}\right)(x, x') \\ \frac{U^{(\ell+1)}(x_1, x_2, x_3, x_4)}{\sigma^4_{w^{(\ell+1)}}} &= \mathcal{U}_\phi\left(K^{(\ell)}, U^{(\ell)}\right)(x_1, x_2, x_3, x_4) \\ &\quad - \mathcal{K}_\phi\left(K^{(\ell)}, U^{(\ell)}\right)(x_{\alpha_1}, x_{\alpha_2})\mathcal{K}_\phi\left(K^{(\ell)}, U^{(\ell)}\right)(x_{\alpha_3}, x_{\alpha_4})\,[3] \end{aligned} \tag{C.4}$$

where the starting point is the first layer ($N^{(0)} \equiv d$)

$$K^{(1)}(x, x') = \frac{\sigma^2_{w^{(1)}}}{N^{(0)}} x \cdot x' \qquad U^{(1)}(x_1, x_2, x_3, x_4) = 0 \qquad (C.5)$$

The important point to note, is that these functional integrals can be reduced to ordinary finite dimensional integrals. For example, for the second layer, denote

$$\mathbf{h} := \begin{pmatrix} h_1 \\ h_2 \end{pmatrix} \qquad \mathbf{K^{(1)}} = \begin{pmatrix} K^{(1)}(x_1, x_1) & K^{(1)}(x_1, x_2) \\ K^{(1)}(x_1, x_2) & K^{(1)}(x_2, x_2) \end{pmatrix} \qquad (C.6)$$

we find for $K^{(2)}$

$$\frac{K^{(2)}(x_1, x_2)}{\sigma^2_{w^{(2)}}} = \int d\mathbf{h} e^{-\frac{1}{2}\mathbf{h}^\mathsf{T}\mathbf{K}_{(1)}^{-1}\mathbf{h}} \phi(h_1) \phi(h_2) \qquad (C.7)$$

and for $U^{(2)}$ we denote

$$\mathbf{h} := \begin{pmatrix} h_1 \\ h_2 \\ h_3 \\ h_4 \end{pmatrix} \qquad \mathbf{K^{(1)}} = \begin{pmatrix} K^{(1)}(x_1, x_1) & K^{(1)}(x_1, x_2) & K^{(1)}(x_1, x_3) & K^{(1)}(x_1, x_4) \\ K^{(1)}(x_1, x_2) & K^{(1)}(x_2, x_2) & K^{(1)}(x_2, x_3) & K^{(1)}(x_2, x_4) \\ K^{(1)}(x_1, x_3) & K^{(1)}(x_2, x_3) & K^{(1)}(x_3, x_3) & K^{(1)}(x_3, x_4) \\ K^{(1)}(x_1, x_4) & K^{(1)}(x_2, x_4) & K^{(1)}(x_3, x_4) & K^{(1)}(x_4, x_4) \end{pmatrix}$$
$$(C.8)$$

so that

$$\mathcal{U}_\phi\left(K^{(1)}, U^{(1)}\right)(x_1, x_2, x_3, x_4) = \int d\mathbf{h} e^{-\frac{1}{2}\mathbf{h}^\mathsf{T}\mathbf{K}_{(1)}^{-1}\mathbf{h}} \phi(h_1) \phi(h_2) \phi(h_3) \phi(h_4) \qquad (C.9)$$

This iterative process can be repeated for an arbitrary number of layers.

# D   FOURTH CUMULANT FOR THRESHOLD POWER-LAW ACTIVATION FUNCTIONS

## D.1   FOURTH CUMULANT FOR ReLU ACTIVATION FUNCTION

The $U$'s appearing in our FWC results can be derived for several activations functions, and in our numerical experiments we use a quadratic activation $\phi(z) = z^2$ and ReLU. Here we give the result for ReLU, which is similar for any other threshold power law activation (see derivation in App. D.2), and give the result for quadratic activation in App. E. For simplicity, in this section we focus on the case of a 2-layer FCN with no biases, input dimension $d$ and $N$ neurons in the hidden layer, such that $\phi_\alpha^i := \phi(w^{(i)} \cdot x_\alpha)$ is the activation at the $i$th hidden unit with input $x_\alpha$ sampled with a uniform measure from $\mathbb{S}_{d-1}(\sqrt{d})$, where $w^{(i)}$ is a vector of weights of the first layer. This can be generalized to the more realistic settings of deeper nets and un-normalized inputs, where in the former the linear kernel $L$ is replaced by the kernel of the layer preceding the output, and the latter amounts to introducing some scaling factors.

For $\phi = $ ReLU, (Cho & Saul, 2009) give a closed form expression for the kernel which corresponds to the GP. Here we find $U$ corresponding to the leading FWC by first finding the fourth moment of the hidden layer $\mu_4 := \langle \phi_1 \phi_2 \phi_3 \phi_4 \rangle$ (see Eq. 9), taking for simplicity $\varsigma_w^2 = 1$

$$\mu_4 = \frac{\sqrt{\det(L^{-1})}}{(2\pi)^2} \int\limits_0^\infty d\mathbf{z} e^{-\frac{1}{2}\mathbf{z}^\mathsf{T} L^{-1}\mathbf{z}} z_1 z_2 z_3 z_4 \qquad (D.1)$$

where $L^{-1}$ above corresponds to the matrix inverse of the $4 \times 4$ matrix with elements $L_{\alpha\beta} = (x_\alpha \cdot x_\beta)/d$ which is the kernel of the previous layer (the linear kernel in the 2-layer case) evaluated on two random points. In App. D.2 we follow the derivation in Moran (1948), which yields (with a slight modification noted therein) the following series in the off-diagonal elements of the matrix $L$

$$\mu_4 = \sum\limits_{\ell,m,n,p,q,r=0}^\infty A_{\ell mnpqr} L_{12}^\ell L_{13}^m L_{14}^n L_{23}^p L_{24}^q L_{34}^r \qquad (D.2)$$

where the coefficients $A_{\ell mnpqr}$ are

$$\frac{(-)^{\ell+m+n+p+q+r} G_{\ell+m+n} G_{\ell+p+q} G_{m+p+r} G_{n+q+r}}{\ell! m! n! p! q! r!} \tag{D.3}$$

For ReLU activation, these $G$'s read

$$G_s^{\text{ReLU}} = \begin{cases} \frac{1}{\sqrt{2\pi}} & s = 0 \\ \frac{-i}{2} & s = 1 \\ 0 & s \geq 3 \text{ and odd} \\ \frac{(-)^k (2k)!}{\sqrt{2\pi} 2^k k!} & s = 2k+2 \quad k = 0, 1, 2, ... \end{cases} \tag{D.4}$$

and similar expressions can be derived for other threshold power-law activations of the form $\phi(z) = \Theta(z) z^\nu$. The series Eq. D.2 is expected to converge for sufficiently large input dimension $d$ since the overlap between random normalized inputs scales as $\mathcal{O}(1/\sqrt{d})$ and consequently $L(x, x') \sim \mathcal{O}(1/\sqrt{d})$ for two random points from the data sets. However, when we sum over $U_{\alpha_1 ... \alpha_4}$ we also have terms with repeating indices and so $L_{\alpha\beta}$'s are equal to 1. The above Taylor expansion diverges whenever the $4 \times 4$ matrix $L_{\alpha\beta} - \delta_{\alpha\beta}$ has eigenvalues larger than 1. Notably this divergence does not reflect a true divergence of $U$, but rather the failure of representing it using the above expansion. Therefore at large $n$, one can opt to neglect elements of $U$ with repeating indices, since there are much fewer of these. Alternatively this can be dealt with by a re-parameterization of the $z$'s leading to a similar but slightly more involved Taylor series.

### D.2 DERIVATION OF THE PREVIOUS SUBSECTION

In this section we derive the expression for the fourth moment $\langle f_1 f_2 f_3 f_4 \rangle$ of a two-layer fully connected network with threshold-power law activations with exponent $\nu$: $\phi(z) = \Theta(z) z^\nu$; $\nu = 0$ corresponds to a step function, $\nu = 1$ corresponds to ReLU, $\nu = 2$ corresponds to ReQU (rectified quadratic unit) and so forth.

When the inputs are normalized to lie on the hypersphere, the matrix $L$ is

$$L = \begin{pmatrix} 1 & L_{12} & L_{13} & L_{14} \\ L_{12} & 1 & L_{23} & L_{24} \\ L_{13} & L_{23} & 1 & L_{34} \\ L_{14} & L_{24} & L_{34} & 1 \end{pmatrix} \tag{D.5}$$

where the off diagonal elements here have $L_{\alpha\beta} = \mathcal{O}\left(1/\sqrt{d}\right)$. We follow the derivation in Ref. Moran (1948), which computes the probability mass of the positive orthant for a quadrivariate Gaussian distribution with covariance matrix $L$:

$$P_+ = \frac{\sqrt{\det(L^{-1})}}{(2\pi)^2} \int_0^\infty d\mathbf{z} \, e^{-\frac{1}{2} \mathbf{z}^\top L^{-1} \mathbf{z}} \tag{D.6}$$

The characteristic function (Fourier transform) of this distribution is

$$\varphi(t_1, t_2, t_3, t_4)$$

$$= \exp\left(-\frac{1}{2} \mathbf{t}^\top L \mathbf{t}\right)$$

$$= \exp\left(-\frac{1}{2} \sum_{\alpha=1}^4 t_\alpha^2\right) \exp\left(-\sum_{\alpha<\beta} L_{\alpha\beta} t_\alpha t_\beta\right)$$

$$= \exp\left(-\frac{1}{2} \sum_{\alpha=1}^4 t_\alpha^2\right) \sum_{\ell,m,n,p,q,r=0}^\infty \frac{(-)^{\ell+m+n+p+q+r} L_{12}^\ell L_{13}^m L_{14}^n L_{23}^p L_{24}^q L_{34}^r}{\ell! m! n! p! q! r!} t_1^{\ell+m+n} t_2^{\ell+p+q} t_3^{m+p+r} t_4^{n+q+r} \tag{D.7}$$

Performing an inverse Fourier transform, we may now write the positive orthant probability as

$$
\begin{aligned}
P_+ &= \frac{1}{(2\pi)^4} \int_{\mathbb{R}_+^4} d\mathbf{z} \int_{\mathbb{R}^4} d\mathbf{t} \; \varphi\left(t_1, t_2, t_3, t_4\right) e^{-i\sum_{\alpha=1}^4 z_\alpha t_\alpha} \\
&= \sum_{\ell,m,n,p,q,r=0}^{\infty} \frac{(-)^{\ell+m+n+p+q+r} L_{12}^\ell L_{13}^m L_{14}^n L_{23}^p L_{24}^q L_{34}^r}{\ell!m!n!p!q!r!} \times \cdots \\
&\quad \times \frac{1}{(2\pi)^4} \int_{\mathbb{R}_+^4} d\mathbf{z} \int_{\mathbb{R}^4} d\mathbf{t} \; e^{\sum_{\alpha=1}^4 \left(-\frac{1}{2}t_\alpha^2 - iz_\alpha t_\alpha\right)} t_1^{\ell+m+n} t_2^{\ell+p+q} t_3^{m+p+r} t_4^{n+q+r} \\
&= \sum_{\ell,m,n,p,q,r=0}^{\infty} A_{\ell mnpqr} L_{12}^\ell L_{13}^m L_{14}^n L_{23}^p L_{24}^q L_{34}^r
\end{aligned}
\tag{D.8}
$$

where the coefficients $A_{\ell mnpqr}$ are

$$
A_{\ell mnpqr} = \frac{(-)^{\ell+m+n+p+q+r} G_{\ell+m+n} G_{\ell+p+q} G_{m+p+r} G_{n+q+r}}{\ell!m!n!p!q!r!}
\tag{D.9}
$$

and the one dimensional integral is

$$
G_s^{(\nu=0)} = \frac{1}{2\pi} \int_0^\infty dz \int_{-\infty}^\infty t^s \exp\left(-\frac{1}{2}t^2 - itz\right) dt
\tag{D.10}
$$

We can evaluate the integral over $t$ to get

$$
G_s^{(\nu=0)} = \frac{1}{(-i)^s (2\pi)^{1/2}} \int_0^\infty \left(\frac{d}{dz}\right)^s e^{-z^2/2} dz
\tag{D.11}
$$

and performing the integral over $z$ yields

$$
G_s^{(\nu=0)} = \begin{cases} \frac{1}{2} & s = 0 \\ 0 & s \text{ even and } s \geq 2 \\ \frac{(2k)!}{i(2\pi)^{1/2} 2^k k!} & s = 2k+1 \quad k = 0, 1, 2, \ldots \end{cases}
\tag{D.12}
$$

We can now obtain the result for any integer $\nu$ by inserting $z^\nu$ inside the $z$ integral:

$$
G_s^{(\nu)} = \frac{1}{2\pi} \int_0^\infty dz \, z^\nu \int_{-\infty}^\infty t^s \exp\left(-\frac{1}{2}t^2 - itz\right) dt = \frac{1}{(-i)^s (2\pi)^{1/2}} \int_0^\infty z^\nu \left(\frac{d}{dz}\right)^s e^{-z^2/2} dz
\tag{D.13}
$$

Using integration by parts we arrive at the result Eq. D.4 reported in the main text

$$
G_s^{\text{ReLU}} = G_s^{(\nu=1)} = \begin{cases} \frac{1}{\sqrt{2\pi}} & s = 0 \\ \frac{-i}{2} & s = 1 \\ 0 & s \geq 3 \text{ and odd} \\ \frac{(-)^k (2k)!}{\sqrt{2\pi} 2^k k!} & s = 2k+2 \quad k = 0, 1, 2, \ldots \end{cases}
\tag{D.14}
$$

Similar expressions can be derived for other threshold power-law activations of the form $\phi(z) = \Theta(z) z^\nu$ for arbitrary integer $\nu$. In a more realistic setting, the inputs $x$ may not be perfectly normalized, in which case the diagonal elements of $L$ are not unity. It amounts to introducing a scaling factor for each of the four $z$'s and makes the expressions a little less neat but poses no real obstacle.

# E  FOURTH CUMULANT FOR QUADRATIC ACTIVATION FUNCTION

For a two-layer network, we may write $U$, the 4th cumulant of the output $f(x) = \sum_{i=1}^{N} a_i \phi(w_i^\mathsf{T} x)$, with $a_i \sim \mathcal{N}(0, \varsigma_a^2/N)$ and $w_i \sim \mathcal{N}(\mathbf{0}, (\varsigma_w^2/d)I)$ for a general activation function $\phi$ as

$$U_{\alpha_1,\alpha_2,\alpha_3,\alpha_4} = \frac{\varsigma_a^4}{N} \left( V_{(\alpha_1,\alpha_2),(\alpha_3,\alpha_4)} + V_{(\alpha_1,\alpha_3),(\alpha_2,\alpha_4)} + V_{(\alpha_1,\alpha_4),(\alpha_2,\alpha_3)} \right) \tag{E.1}$$

with

$$V_{(\alpha_1,\alpha_2),(\alpha_3,\alpha_4)} = \langle \phi^{\alpha_1} \phi^{\alpha_2} \phi^{\alpha_3} \phi^{\alpha_4} \rangle_w - \langle \phi^{\alpha_1} \phi^{\alpha_2} \rangle_w \langle \phi^{\alpha_3} \phi^{\alpha_4} \rangle_w \tag{E.2}$$

For the case of a quadratic activation function $\phi(z) = z^2$ the $V$'s read

$$V_{(\alpha_1,\alpha_2),(\alpha_3,\alpha_4)} = 2 \left\{ L_{11} L_{33} (L_{24})^2 + L_{11} L_{44} (L_{23})^2 + L_{22} L_{33} (L_{14})^2 + L_{22} L_{44} (L_{13})^2 \right\} + ...$$
$$4 \left\{ (L_{13})^2 (L_{24})^2 + (L_{14})^2 (L_{23})^2 \right\} + 8 \left( L_{11} L_{23} L_{34} L_{24} + L_{22} L_{34} L_{14} L_{13} + L_{33} L_{12} L_{14} L_{24} + L_{44} L_{12} L_{13} L_{23} \right) + ...$$
$$16 \left( L_{12} L_{13} L_{24} L_{34} + L_{12} L_{14} L_{23} L_{34} + L_{13} L_{14} L_{23} L_{24} \right) \tag{E.3}$$

where the linear kernel from the first layer is $L(x, x') = \frac{\varsigma_w^2}{d} x \cdot x'$. Notice that we distinguish between the scaled and non-scaled variances:

$$\sigma_a^2 = \frac{\varsigma_a^2}{N}; \qquad \sigma_w^2 = \frac{\varsigma_w^2}{d} \tag{E.4}$$

These formulae were used when comparing the outputs of the empirical two-layer network with our FWC theory Eq. 11. One can generalize them straightforwardly to a network with $M$ layers by recursively computing $K^{(M-1)}$ the kernel in the $(M-1)$th layer (see e.g. Cho & Saul (2009)), and replacing $L$ with $K^{(M-1)}$.

# F  AUTO-CORRELATION TIME AND ERGODICITY

As mentioned in the main text, the network outputs $\bar{f}_{\mathrm{DNN}}(x_*)$ are a result of averaging across many realizations (seeds) of initial conditions and the noisy training dynamics, and across time (epochs) after the training loss levels off. Our NNSP correspondence relies on the fact that our stochastic training dynamics are ergodic, namely that averages across time equal ensemble averages. Actually, for our purposes it suffices that the dynamics are *ergodic in the mean*, namely that the time-average estimate of the mean obtained from a single sample realization of the process converges in both the mean and in the mean-square sense to the ensemble mean:

$$\lim_{\tilde{T} \to \infty} \mathbb{E} \left[ \langle f^{\mathrm{DNN}}(x_*; t) \rangle_{\tilde{T}} - \mu(x_*) \right] = 0$$
$$\lim_{\tilde{T} \to \infty} \mathbb{E} \left[ \left( \langle f^{\mathrm{DNN}}(x_*; t) \rangle_{\tilde{T}} - \mu(x_*) \right)^2 \right] = 0 \tag{F.1}$$

where $\mu(x_*)$ is the ensemble mean on the test point $x_*$ and the time-average estimate of the mean over a time window $\tilde{T}$ is

$$\langle f^{\mathrm{DNN}}(x_*; t) \rangle_{\tilde{T}} := \frac{1}{\tilde{T}} \int_0^{\tilde{T}} f^{\mathrm{DNN}}(x_*; t) dt \approx \frac{1}{\tilde{T}} \sum_{t_j=0}^{t_j=\tilde{T}} f^{\mathrm{DNN}}(x_*; t_j) \tag{F.2}$$

This is hard to prove rigorously but we can do a numerical consistency check using the following procedure: Consider the time series of the network output on the test point $x_*$ for the $i$'th realization as a row vector and stack these row vectors for all different realizations into a matrix $F$, such that $F_{ij} = f_i^{\mathrm{DNN}}(x_*; t_j)$. (1) Divide the time series data in the matrix $F$ into non-overlapping sub-matrices, each of dimension $n_{\mathrm{seeds}} \times n_{\mathrm{epochs}}$. (2) For each of these sub-matrices, find $\hat{f}(x_*)$ i.e. the empirical dynamical average across that time window and across the chosen seeds; (2) Find

the empirical variance $\sigma^2_{\text{emp}}(x_*)$ across these $\hat{f}(x_*)$; (4) Repeat (1)-(3) for other combinations of $n_{\text{epochs}}, n_{\text{seeds}}$. If ergodicity holds, we should expect to see the following relation

$$\sigma^2_{\text{emp}}(x_*) = \sigma^2_m \frac{\tau}{n_{\text{epochs}} n_{\text{seeds}}} \tag{F.3}$$

where $\tau$ is the auto-correlation time of the outputs and $\sigma^2_m$ is the macroscopic variance. The results of this procedure are shown in Fig. F.1, where we plot on a log-log scale the empirical variance $\sigma^2_{\text{emp}}$ vs. the number of epochs $n_{\text{epochs}}$ used for time averaging in each set (and using all 500 seeds in this case). Performing a linear fit on the average across test points (black x's in the figure) yields a slope of approximately $-1$, which is strong evidence for ergodic dynamics.

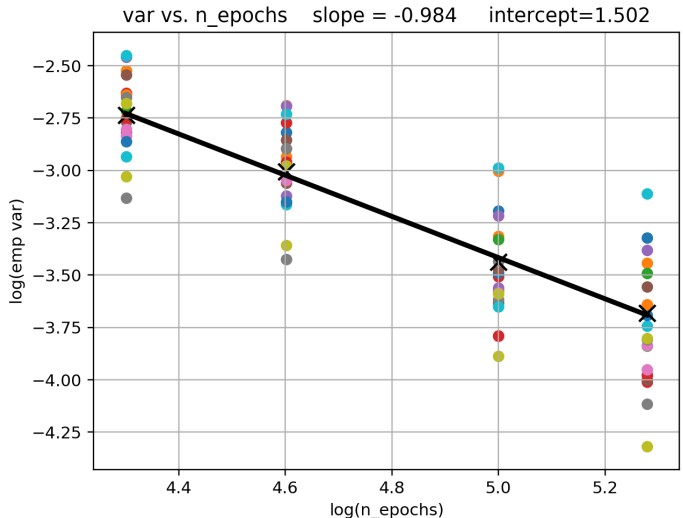

**Figure F.1: Ergodicity check.** Empirical variance $\sigma^2_{\text{emp}}(x_*)$ vs. the number of epochs used for time averaging on a (base 10) log-log scale, with $dt = 0.003$ and $N = 200$. The colored circles represent different test points $x_*$ and the black x's are averages across these.

## G    NUMERICAL EXPERIMENT DETAILS

### G.1    FCN EXPERIMENT DETAILS

We trained a 2-layer FCN on a quadratic target $y(x) = x^\mathsf{T} A x$ where the $x$'s are sampled with a uniform measure from the hyper-sphere $\mathbb{S}_{d-1}(\sqrt{d})$, with $d = 16$ and the matrix elements are sampled as $A_{ij} \sim \mathcal{N}(0, 1)$ and fixed for all $x$'s. For both activation functions, we used a training noise level of $\sigma^2 = 0.2$, training set of size $n = 110$ and a weight decay of the first layer $\gamma_w = 0.05$. Notice that for any activation $\phi$, $K$ scales linearly with $\varsigma^2_a = \sigma^2_a N = (T/\gamma_a) \cdot N$, thus in order to keep $K$ constant as we vary $N$ we need to scale the weight decay of the last layer as $\gamma_a \sim \mathcal{O}(N)$. This is done in order to keep the prior distribution in accord with the typical values of the target as $N$ varies, so that the comparison is fair.

We ran each experiment for $2 \cdot 10^6$ epochs, which includes the time it takes for the training loss to level off, which is usually on the order of $10^4$ epochs. In the main text we showed GP and FWC results for a learning rate of $dt = 0.001$. Here we report in Fig. G.1 the results using $dt \in \{0.003, 0.001, 0.0005\}$. For a learning rate of $dt = 0.003$ and width $N \geq 1000$ the dynamics become unstable and strongly oscillate, thus the general trend is broken, as seen in the blue markers in Fig. G.1. The dynamics with the smaller learning rates are stable, and we see that there is a convergence to very similar values up to an expected statistical error.

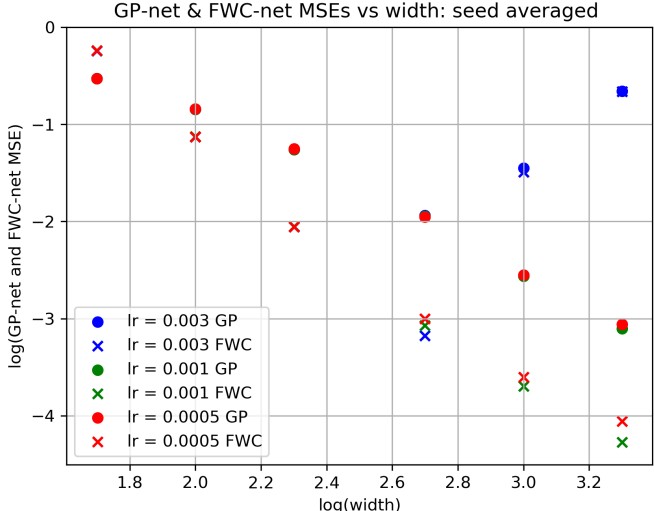

**Figure G.1: Regression task with fully connected network: (un-normalized) MSE vs. width on log-log scale (base 10) for quadratic activation and different leaning rates.** The learning rates $dt = 0.001, 0.0005$ converge to very similar values (recall this is a log scale), demonstrating that the learning rate is sufficiently small so that the discrete-time dynamics is a good approximation of the continuous-time dynamics. For a learning rate of $dt = 0.003$ (blue) and width $N \geq 1000$ the dynamics become unstable, thus the general trend is broken, so one cannot take the $dt$ to be too large.

### G.2  CNN EXPERIMENT DETAILS AND ADDITIONAL SETTINGS

The CNN experiment reported in the main text was carried as follows.

**Dataset:** In the main text Fig. 3 we used a random sample of 10 train-points and 2000 test points from the CIFAR10 dataset, and in App. H we report results on 1000 train-points and 1000 test points, balanced in terms of labels. To use MSE loss, the ten categorical labels were one-hot encoded into vector of zeros and one.

**Architecture:** we used 6 convolutional layers with ReLU non-linearity, kernel of size $5 \times 5$, stride of 1, no-padding, no-pooling. The number of input channels was 3 for the input layer and $C$ for the subsequent 5 CNN layers. We then vectorized the outputs of the final layer and fed it into an ReLU activated fully-connected layer with $25C$ outputs, which were fed into a linear layer with 10 outputs corresponding to the ten categories. The loss we used was MSE loss.

**Training:** Training was carried using full-batch SGD (GD) at varying learning-rates around $5 \cdot 10^{-4}$, Gaussian white noise was added to the gradients to generate $\sigma^2 = 0.2$ in the NNGP-correspondence, layer-dependant weight decay and bias decay which implies a (normalized by width) weight variance and bias variance of $\sigma_w^2 = 2$ and $\sigma_b^2 = 1$ respectively, when trained with no-data. During training we saved, every 1000 epochs, the outputs of the CNN on every test point. We note in passing that the standard deviation of the test outputs around their training-time-averaged value was about 0.1 per CNN output. Training was carried for around half a million epochs which enabled us to reach a statistical error of about $2 \cdot 10^{-4}$, in estimating the Mean-Squared-Discrepancy between the training-time-averaged CNN outputs and our NNGP predictions. Notably our best agreement between the DNN and GP occurred at 112 channels where the MSE was about $7 \cdot 10^{-3}$. Notably the variance of the CNN (the average of its outputs squared) with no data, was about 25.

**Statistics.** To train our CNN within the regime of the NNSP correspondence, sufficient training time (namely, epochs) was needed to get estimates of the average outputs $\bar{f}_E(x_\alpha) = \bar{f}(x_\alpha) + \delta f_\alpha$ since

the estimators' fluctuations, $\delta f_\alpha$, scale as $(\tau/t_{\text{training}})^{-1/2}$, where $\tau$ is an auto-correlation time scale. Notably, apart from just random noise when estimating the relative MSE between the averaged CNN outputs and the GP, a bias term appears equal to the variance of $\delta f_\alpha$ averaged over all $\alpha$'s as indeed

$$\sum_{\alpha=1}^{n_{\text{test}}} (\bar{f}_E(x_\alpha) - f_{GP}(x_\alpha))^2 = \sum_{\alpha=1}^{n_{\text{test}}} (\bar{f}(x_\alpha) - f_{GP}(x_\alpha))^2 - 2 \sum_{\alpha=1}^{n_{\text{test}}} (\bar{f}_E(x_\alpha) - f_{GP}(x_\alpha))\delta f_\alpha + \sum_{\alpha=1}^{n_{\text{test}}} (\delta f_\alpha)^2$$

(G.1)

In all our experiments this bias was the dominant source of statistical error. One can estimate it roughly given the number of uncorrelated samples taken into $\bar{f}_E(x_\alpha)$ and correct the estimator. We did not do so in the main text to make the data analysis more transparent. Since the relative MSEs go down to $7 \cdot 10^{-3}$ and the fluctuations of the outputs quantified by $\Sigma_\alpha = (\delta f_\alpha)^2$ are of the order $0.1^2$, the amount of uncorrelated samples of CNN outputs we require should be much larger than $0.1^2/(7 \cdot 10^{-3}) \approx 1.43$. To estimate this bias in practice we repeated the experiment with 3-7 different initialization seeds and deduced the bias from the variance of the results. For comparison with NNGP (our $DNN - GP$ plots) the error bars were proportional to the variance of $\delta f_\alpha$. For comparison with the target, we took much larger error bars equal to the uncertainty in estimating the expected loss from a test set of size 1000. These latter error bars where estimated empirically by measuring the variance across ten smaller test sets of size 100.

Lastly we discarded the initial "burn-in" epochs, where the network has not yet reached equilibrium. We took this burn-in time to be the time it takes the train-loss to reach within $5\%$ of its stationary value at large times. We estimated the stationary values by waiting until the DNNs train loss remained constant (up to trends much smaller than the fluctuations) for about $5 \cdot 10^5$ epochs. This also coincided well with having more or less stationary test loss.

**Learning rate.** To be in the regime of the NNSP correspondence, the learning rate must be taken small enough such that discrepancy resulting from having discretization correction to the continuum Langevin dynamics falls well below those coming from finite-width. We find that higher $C$ require lower learning rates, potentially due to the weight decay term being large at large width. In Fig. G.2. we report the relative MSE between the NNGP and CNN at learning rates of $0.002, 0.001, 0.0005$ and $C = 48$ showing good convergence already at $0.001$. Following this we used learning rates of $0.0005$ for $C \leq 48$ and $0.00025$ for $C > 48$, in the main figure.

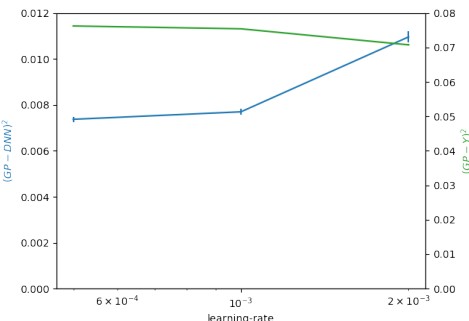

**Figure G.2:** MSE between our CNN with $C = 48$ and its NNGP as a function of three learning rates.

**Comparison with the NNGP.** Following Novak et al. (2018), we obtained the Kernel of our CNN. Notably, since we did not have pooling layers this can be done straightforwardly without any approximations. The NNGP predictions were then obtained in a standard manner (Rasmussen & Williams, 2005).

## H  FURTHER NUMERICAL RESULTS ON CNNS

Here we report two additional numerical results following the CNN experiment we carried (for details see App. G). Fig. H.3b is the same as Fig. H.3a apart from the fact that we subtracted our estimate of the statistical bias of our MSE estimator described in App. G.

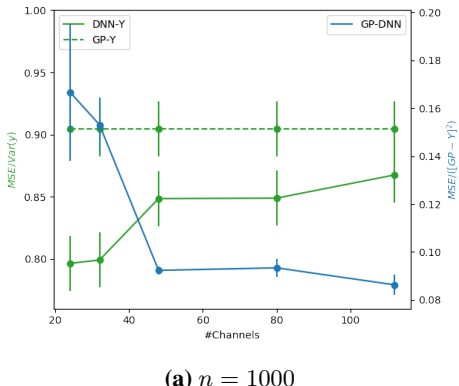 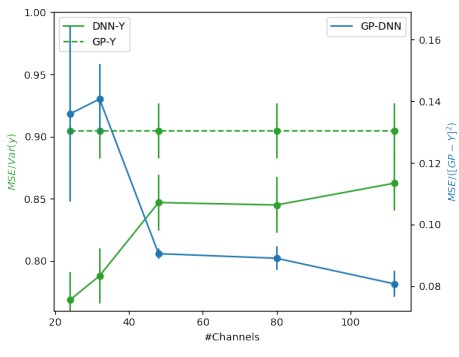

**(a)** $n = 1000$             **(b)** subtract bias of MSE estimator

**Figure H.3: CNNs trained on CIFAR10 in the regime of the NNSP correspondence compared with NNGPs** MSE test loss normalized by target variance of a deep CNN (solid green) and its associated NNGP (dashed green) along with the MSE between the NNGP's predictions and CNN outputs normalized by the NNGP's MSE test loss (solid blue, and on a different scale). We used balanced training and test sets of size 1000 each. For the largest number of channels we reached, the slope of the discrepancy between the CNN's GP and the trained DNN on the log-log scale was $-1.77$, placing us close to the perturbative regime where a slope of $-2.0$ is expected. Error bars here reflect statistical errors related only to output averaging and not due to the random choice of a test-set. The performance *deteriorates* at large $N = \#\text{Channels}$ as the NNSP associated with the CNN approaches an NNGP.

Concerning the experiment with 10 training points. Here we used the same CNN as in the previous experiment. The noise level was again the same and led to an effective $\sigma^2 = 0.1$ for the GP. The weight decay on the biases was taken to be ten times larger leading to $\sigma_b^2 = 0.1$ instead of $\sigma_b = 1.0$ as before. For $C \leq 80$ we used a learning rate of $dt = 5 \cdot 10^{-5}$ after verifying that reducing it further had no appreciable effect. For $C \leq 80$ we used $dt = 2.5 \cdot 10^{-5}$. For $c \leq 80$ we used $6 \cdot 10^{+5}$ training epochs and we averaged over 4 different initialization seeds. For $C > 80$ we used between $10 - 16$ different initialization seeds. We reduced the aforementioned statistical bias in estimating the MSE from all our MSEs. This bias, equal to the variance of the averaged outputs, was estimated based on our different seeds. The error bars equal this estimated variance which was the dominant source of error.

## I  THE FOURTH CUMULANT CAN DIFFERENTIATE CNNS FROM LCNS

Here we show that while the NNGP kernel $K$ of a CNN without pooling cannot distinguish a CNN from an LCN, the fourth cumulant, $U$, can. For simplicity let us consider the simplest CNN without pooling consisting of the following parts: (1) A 1D image with one color/channel ($X_i$) as input $i \in \{0, \ldots, L-1\}$; (2) A single convolutional layer with some activation $\phi$ acting with stride 1 and no-padding using the conv-kernel $T_x^c$ where $c \in \{1, \ldots, C\}$ is a channel number index and $x \in \{0, \ldots, 2l\}$ is the relative position in the image. Notably, in an LCN this conv-kernel will receive an additional dependence on $\tilde{x}$, the location on $X_i$ on which the kernel acts. (3) A vectorizing operation taking the $C$ outputs of each convolutional around a point $\tilde{x} \in \{l, \ldots, L-l\}$, into a single index $y \in \{0, \ldots, C(L-2l)\}$. (4) A linear fully connected layer with weights $W_{c\tilde{x}}^o$ where $o \in \{0, \ldots, \#\text{outputs}\}$ are the output indices.

Consider first the NNGP of such a random DNN with weights chosen according to some iid Gaussian distribution $P_0(w)$, with $w$ including both $W_{c\tilde{x}}^o$ and $T_x^c$. Denoting by $z^o(x)$ the $o$'th output of the CNN, for an input $x$ we have (where we denote in this section $\langle \cdots \rangle := \langle \cdots \rangle_{P_0(w)}$)

$$K^{oo'}(x, x') \equiv \langle z^o(x) z^{o'}(x') \rangle = \delta_{oo'} \sum_{c,c',\tilde{x},\tilde{x}'} \langle W_{c\tilde{x}}^o W_{c'\tilde{x}'}^{o'} \rangle \langle \phi(T_x^c(\tilde{x}) X_{x+\tilde{x}-l}) \phi(T_x^{c'}(\tilde{x}') X_{x'+\tilde{x}'-l}) \rangle$$

$$(\text{I.1})$$

The NNGP kernel of an LCN is the same as that of a CNN. This stems from the fact that $\langle W^o_{c\tilde{x}} W^o_{c'\tilde{x}'} \rangle$ yields a Kronecker delta function on the $\tilde{x}, \tilde{x}'$ indices. Consequently, the difference between LCN and CNN, which amounts to whether $T^c_x(\tilde{x})$ is the same (CNN) or a different (LCN) random variable than $T^c_{x' \neq x}(\tilde{x}')$, becomes irrelevant as the these two are never averaged together.

For simplicity, we turn to the fourth cumulant of the same output, given by

$$\langle z^o(x_1) \cdots z^o(x_4) \rangle - \langle z^o(x_\alpha) z^o(x_\beta) \rangle \langle z^o(x_\gamma) z^o(x_\delta) \rangle [3] = \langle z^o(x_1) \cdots z^o(x_4) \rangle - K(x_\alpha, x_\beta) K(x_\gamma, x_\delta)[3] \tag{I.2}$$

with the second term on the LHS implying all pair-wise averages of $z^o(x_1)..z^o(x_4)$. Note that the first term on the LHS is not directly related to the kernel, thus it has a chance of differentiating a CNN from an LCN. Explicitly, it reads

$$\sum_{c_1..c_4 \tilde{x}_1..\tilde{x}_4} \langle W^o_{c_1\tilde{x}_1} \cdots W^o_{c_4\tilde{x}'_4} \rangle \langle \phi(T^{c_1}_{x_1}(\tilde{x}_1) X_{x_1+\tilde{x}_1-l}) \cdots \phi(T^{c_4}_{x_4}(\tilde{x}_4) X_{x_4+\tilde{x}'_4-l}) \rangle \tag{I.3}$$

The average over the four $W$'s yields non-zero terms of the type $W^o_{c\tilde{x}} W^o_{c\tilde{x}} W^o_{c'\tilde{x}'} W^o_{c'\tilde{x}'}$ with either $\tilde{x} = \tilde{x}'$ (type 1), $\tilde{x} \neq \tilde{x}'$ and $c \neq c'$ (type 2), or $\tilde{x} \neq \tilde{x}'$ and $c = c'$ (type 3).

The type 1 contribution cannot differentiate an LCN form a CNN since, as in the NNGP case, they always involve only one $\tilde{x}$. The type 2 contribution also cannot differentiate since it yields

$$\sum_{c \neq c'; \tilde{x} \neq \tilde{x}'} \langle W^o_{c\tilde{x}} W^o_{c\tilde{x}} \rangle \langle W^o_{c'\tilde{x}'} W^o_{c'\tilde{x}'} \rangle \langle \phi(T^c_x(\tilde{x}) X_{x+\tilde{x}-l}) \phi(T^c_x(\tilde{x}) X_{x+\tilde{x}-l}) \phi(T^{c'}_{x'}(\tilde{x}') X_{x'+\tilde{x}'-l}) \phi(T^{c'}_{x'}(\tilde{x}') X_{x'+\tilde{x}'-l}) \rangle \tag{I.4}$$

Examining the average involving the four $T$'s, one finds that since $T^c_x(\tilde{x})$ is uncorrelated with $T^{c'}_{x'}(\tilde{x}')$ for both LCNs and CNNs, it splits into

$$\sum_{c \neq c'; \tilde{x} \neq \tilde{x}'} \langle W^o_{c\tilde{x}} W^o_{c\tilde{x}} \rangle \langle W^o_{c'\tilde{x}'} W^o_{c'\tilde{x}'} \rangle \langle \phi(T^c_x(\tilde{x}) X_{x+\tilde{x}-l}) \phi(T^c_x(\tilde{x}) X_{x+\tilde{x}-l}) \rangle \langle \phi(T^{c'}_{x'}(\tilde{x}') X_{x'+\tilde{x}'-l}) \phi(T^{c'}_{x'}(\tilde{x}') X_{x'+\tilde{x}'-l}) \rangle \tag{I.5}$$

where as in the NNGP, two $T$'s with different $\tilde{x}$ are never averaged together and we only get a contribution proportional to products of two $K$'s. We note in passing that these type 2 terms yield a contribution that largely cancels that of $K(x_\alpha, x_\beta) K(x_\gamma, x_\delta)[3]$, apart from a "diagonal" contribution ($\tilde{x} = \tilde{x}'$).

We turn our attention to the type 3 term given by

$$\sum_{c; \tilde{x} \neq \tilde{x}'} \langle W^o_{c\tilde{x}} W^o_{c\tilde{x}} \rangle \langle W^o_{c\tilde{x}'} W^o_{c\tilde{x}'} \rangle \langle \phi(T^c_x(\tilde{x}) X_{x+\tilde{x}-l}) \phi(T^c_x(\tilde{x}) X_{x+\tilde{x}-l}) \phi(T^c_{x'}(\tilde{x}') X_{x'+\tilde{x}'-l}) \phi(T^c_{x'}(\tilde{x}') X_{x'+\tilde{x}'-l}) \rangle \tag{I.6}$$

Examining the average involving the four $T$'s, one now finds a sharp difference between an LCN and a CNN. For an LCN, this average would split into a product of two $K$'s since $T^c_x(\tilde{x})$ would be uncorrelated with $T^c_x(\tilde{x}')$. For a CNN however, $T^c_x(\tilde{x})$ is the same random variable as $T^c_x(\tilde{x}')$ and therefore the average does not split giving rise to a distinct contribution that differentiates a CNN from an LCN. Notably, it is small by a factor of $1/C$ owing to the fact that it contains a redundant summation over one $c$-index while the averages over the four $W$'s contain a $1/C^2$ factor when properly normalized.

## J  CORRECTIONS TO EK

Here we derive finite-$N$ correction to the Equivalent Kernel result. Using the tools developed by Cohen et al. (2019), the replicated partition function relevant for estimating the predictions of the network ($f(x_*)$) averaged ($\langle \cdots \rangle_n$) over all draws of datasets of size $n'$ with $n'$ taken from a Poisson distribution with mean $n$ is given by

$$Z_n = \int \mathcal{D} f e^{-S_{\mathrm{GP}}[f] - \frac{n}{2\sigma^2} \int d\mu_x (f(x) - y(x))^2} (1 + S_U[f]) + \mathcal{O}(1/N^2) \tag{J.1}$$

with $S_{\mathrm{GP}}[f]$ and $S_U[f]$ given in Eq. 8. We comment that the above expression is only valid for obtaining the leading order asymptotics in $n$. Enabling generic $n$ requires introducing replicas explicitly (see Cohen et al. (2019)). Notably, the above expression coincides with that used for

a finite dataset, with two main differences: all the sums over the training set have been replaced by integrals with respect to the measure, $\mu_x$, from which data points are drawn. Furthermore $\sigma^2$ is now accompanied by $n$. Following this, all the diagrammatic and combinatorial aspects shown in the derivation for a finite dataset hold here as well. For instance, let us examine a specific contribution coming from the quartic term in $H[f]$: $U_{x_1..x_4} K^{-1}_{x_1 x_1'} \cdots K^{-1}_{x_4 x_4'} f(x_1') \cdots f(x_4')$, and from the diagram/Wick-contraction where we take the expectation value of $3$ out of the $4$ $f$'s in this quartic term, to arrive at an expression which is ultimately cubic in the targets $y$

$$U_{x_1,x_2,x_3,x_4} K^{-1}_{x_1 x_1'} \langle f(x_1') \rangle_\infty K^{-1}_{x_2 x_2'} \langle f(x_2') \rangle_\infty K^{-1}_{x_3 x_3'} \langle f(x_3') \rangle_\infty K^{-1}_{x_4 x_4'} \Sigma_\infty(x_4', x_*) \qquad \text{(J.2)}$$

where we recall that $\langle f(x) \rangle_\infty = K_{xx'} \tilde{K}^{-1}_{x'x''} y(x'')$ and $\Sigma_\infty(x_1, x_2) = K_{x_1,x_2} - K_{x_1,x'} \tilde{K}^{-1}_{x',x''} K_{x'',x_2}$ being the posterior covariance in the EK limit, where $\tilde{K}_{xx'} f(x') = K_{xx'} f(x') + (\sigma^2/n) f(x)$. Using the fact that $K^{-1}_{xx'} K_{x'x''}$ gives a delta function w.r.t. the measure, the integrals against $K^{-1}_{x_\alpha x_\alpha'}$ can be easily carried out yielding

$$\left( U_{x_1,x_2,x_3,x_*} - U_{x_1,x_2,x_3,x_4} \tilde{K}^{-1}_{x_4,x_4'} K_{x_4',x_*} \right) \tilde{K}^{-1}_{x_1,x_1'} \tilde{K}^{-1}_{x_2,x_2'} \tilde{K}^{-1}_{x_3,x_3'} y(x_1') y(x_2') y(x_3') \qquad \text{(J.3)}$$

Introducing the discrepancy operator $\tilde{\delta}_{xx''} := \delta_{xx''} - K_{xx'} \tilde{K}^{-1}_{x'x''} = \frac{\sigma^2}{n} \tilde{K}^{-1}_{xx''}$, we can write a more compact expression

$$\left( \frac{n}{\sigma^2} \right)^3 \tilde{\delta}_{x_*,x_4} U_{x_1,x_2,x_3,x_4} \tilde{\delta}_{x_1,x_1'} \tilde{\delta}_{x_2,x_2'} \tilde{\delta}_{x_3,x_3'} y(x_1')(x_2') y(x_3') \qquad \text{(J.4)}$$

This with the additional $1/4!$ factor times the combinatorial factor of $4$ related to choosing the "partner" of $f(x_*)$ in the Wick contraction, yields an overall factor of $1/6$ as in the main text, Eq. 14. The other term therein, which is linear in $y$, is a result of following similar steps with the $\bar{f} \Sigma \Sigma_*$ contributions that do not get canceled by the quadratic part in $H[f]$.

