# OpenReview forum: "Predicting the Outputs of Finite Networks Trained with Noisy Gradients"
_ICLR.cc/2021/Conference — Reject_

### Official Review · AnonReviewer1 · 2020-10-27
**Interesting theory on finite width DNNs but not clearly explained**

**Rating:** 4
**Confidence:** 2

**Review:**

It has been show that practical performance of finite width DNNs deviates from the infinite limiting cases of both the NTK and GP. The reason behind this is still not well understood. This papers develops theory trying to explain this by showing the finite width correction (FWC) for DNNs. The results look very interesting. However, as an educational guest on this topic, I find the paper written in a way that causes a lot of confusions, making me not fully understand the results.

First of all, the paper is talking about the FWC. Although it has been used in previous work and I can get some sense of the meaning, I think a formal definition is needed in order to make the paper more readable.

Regarding the technical details:

1. What is Df in eq.4? And why can it be eliminated? And it is not clear to me about the sentence below eq.4: why the rhs of eq.4 equals the kernel of the NNGP only in highly over-parameterized DNNs?
2. I am not sure how eq.5 is obtained?
3. I think the description below eq.5 can be made more formal because this gives the definition of NNSP. Based on the current description, I still cannot get why NNSP is special? Is it still a GP and why?
4. Second line below section 3.1, it says e^{-L[f]/2\sigma^2} is independent of the DNNs, but why? I think f corresponds to the DNN?
5. In Section 3, the paper derives complicated formulations for the posterior mean and variance. I am not sure how these results are useful? It seems to say that these results describe how the finite width DNNs are deviated from the NTK and NNGP, but by looking at these formulas, it is hard for me to figure out why NNSP is better than NTK and NNGP?
6. Paragraph above Figure 1: it says "the above cubic term would lose its explicit dependence on n", but I think there is still a quadratic term, which still depends on n. How can it say that the FWC is negligible in the large n regime?

Overall, I found this paper addressing an important and interesting problem, but the current presentation cannot convince me a pass.

---

> ### Author Response · Authors · 2020-11-24
> **Response to Reviewer 1 - part 1**
>
> Dear reviewer,
>
> Thank you for your review. We will answer your questions in order of appearance.
> 1. **Clarity of presentation**:
>
> (a) **Formal definition of FWC** -
> The FWC is the correction to GP prior due to finite width. We derived its leading term in $1/N$: at the level of correction to the prior over functions, this is given in Eqs. 7-9, and at the level of correction to the predictive mean and variance, this is given in Eqs. 10-12.
> As stated in the paper, it is well known that as $N \to \infty$, the distribution in function space of the NN output is Gaussian, i.e. it becomes a Gaussian process governed by the kernel $K$. Thus there are simple formulas for the predictive mean and variance as a function of the training data and the kernel $K$, which in the NNGP correspondence depends on the NN architecture (activation function, number of layers, certain hyper-parameters etc.). These are given in Eq. 6.
> Our FWC is a leading order correction in one over the width ($1/N$) to these well-known results from the theory of GPs. Viewing GPs as a linear inference tool (its predictions are a linear function of the target), our finite width corrections contain the leading order non-linear corrections (they contained terms which are cubic in the target).
>
> (b) Could you elaborate on what other specific points caused confusion?
>
> 2. **Technical details**:
> (a) **Eq. 4 and the following sentence** -
>  $\int\mathcal{D}f$ means integration over function space. We introduced a delta function in function space that enforces the DNN output to be $z_{w}=f$, thus exchanging $\int\mathcal{D}f$ for $\int dw$.
> Technically, such a delta function is used as
> $
> \int \mathcal{D}f \delta[f - f_0] f(x) f(y) f(z) \cdots =
> f_0(x) f_0(y) f_0(z) \cdots
> $.
> It is a formal tool that is part of the path integral approach to stochastic and/or quantum systems (For a review see Schulman 2012 in our bibliography).
> The fact that the RHS of Eq. 4 equals the kernel of the corresponding NNGP can be understood in several steps. The first equality simply uses the definition of the prior ($P_0[f]$) in terms of the weights as well as the above formula to trade $z_w(x)$ with $f(x)$. The second equality amounts to noticing that $\int \mathcal{D} f \delta[f-f_0]=1$. The final expression on the r.h.s. is the very definition of $K(x,x')$.
>
> (b) **Eq. 5** -
> Here we first recall the well-known result of Cho \& Saul 2009 (also Neal 1996), that DNNs at $N\rightarrow \infty$ correspond to GPs with a covariance $K(x,x')$ related to the correlation of the outputs on the ensemble of random Gaussian iid weights. Second we notices that the r.h.s. of Eq. 5, defines a GP whose covariance is that $K(x,x')$ (See also Schulman 2012 for how to obtain the average of $f(x)f(x')$ under the path-integral ($\int \mathcal{D} f...$) and showing that it also yields $K(x,x')$). Hence the two describe a Gaussian process with the same covariance and hence they coincide.
> The $O\left(1/N\right)$ correction is our leading FWC, the scaling of which originates from the properties of cumulants appearing in the Edgeworth expansion, explained in section 3.

---

> > ### Author Response · Authors · 2020-11-24
> > **Response to Reviewer 1 - part 2**
> >
> > (c) **Definition of NNSP** -
> > We accept that the formal definition of the NNSP could be made more clear and will revise this in the text.
> > The NNSP is \textbf{not} a GP as it involves a higher-order cumulant $U$, whereas a GP is completely defined by
> > its kernel $K$ (and its mean which is usually taken to be zero). The NNSP is the stochastic process, that to leading order in $1/N$ depends only on $K$ and $U$ (higher cumulants are $O(1/N^2)$ or smaller). It is worth considering because it gives more accurate predictions of the network output relative to the corresponding GP.
> >
> > (d) **$e^{-\mathcal{L}[f]/2\sigma^{2}}$ is independent of the DNNs** -
> > This is seen in equations 2 and 3: the DNN architecture is encoded in $z_{w}$, but when we integrate over weights (as in Eq. 3) we get
> > $$
> > \int dw \delta[f-z_w] e^{-\frac{1}{2\sigma^{2}} \mathcal{L} \left(z_{w}  \right)} =
> >     e^{-\frac{1}{2\sigma^2} \mathcal{L} \left[f\right]}
> > $$
> > which is independent of the DNN architecture and only depends on the data.
> > In fact, the only reminder that $f$ reflects the output of the NN now comes from the prior term where, up to $O(1/N^2)$, all the architecture is encoded into $K$ and $U$.
> >
> > (e) **Usefulness of the FWC analytical results**
> > these results are useful for several reasons:
> >
> > i. They constitute a closed-form expression that gives a better approximation than NNGP for the output of a finite width NN, as validated in Figure 1.
> >
> > ii. They are more easily interpretable than an actual NN, which is a black box. Specifically,
> > it is a cubic + linear function of the training targets $y_{\alpha}$ where the coefficients are given in terms of the second and fourth cumulants $K,U$, which can be computed analytically for several known activation functions (for instance, we provide an expression for $U$ for ReLU activation in the Appendix).
> >
> > iii. Moreover, we can write it in terms of the discrepancy (RMSE) of the corresponding GP predictions, which gives additional insight, e.g. that the FWC will be small whenever GP has a small error (see Eq 14 and the subsequent discussion). This is in fact one of our central observations-- if the GP is very good at predicting the target the DNN converges to its infinite-width limit more rapidly. We believe a similar result would hold for DNNs trained in the NTK-regime, but this is outside the scope of the current manuscript.
> >
> > (f) **Dependence on $n$ in Eq. 14** -
> > the scaling with $n$ of this second term in Eq. 14 is indeed less obvious. The numerical results suggest that $\tilde{\delta}_{x_2 x_3}$
> > also scales as $\sigma^2/n$, so that the whole expression in the {...} has no scaling with $n$. This is now clarified in the text.

---

### Official Review · AnonReviewer3 · 2020-10-29
**Interesting and new correspondence between neural networks and Gaussian kernels**

**Rating:** 6
**Confidence:** 4

**Review:**

This paper studies equivalence between neural networks and Gaussian process. The authors make three main contributions: (1) they prove that infinite-width DNNs trained with noisy gradients are corresponded to neural network Gaussian process (NNGP), not neural tangent kernel (NTK); (2) they prove that the finite width correction (FWC) for DNNs with arbitrary activation functions and depth has a general analytical form which can help predict the outputs of empirical finite networks with high accuracy; and (3) they show how these FWCs can improve the performance of finite convolutional neural networks (CNNs) on image classification tasks.

Overall, I vote for accepting. The paper establishes solid correspondence between neural networks and Gaussian process which may have a real impact to applications. The authors also give empirical results to support their arguments.

Pros:

+ The paper takes one of the most important problem in deep learning theory – the correspondence between neural networks and Gaussian kernels.

+ The paper makes real contributions compared with existing works on NNGP (in NNGP, weights were determined by the distribution of the DNN weights at initialization; and in this paper, the weights are sampled across the stochastic training dynamics).

+ Give results for finite-width networks, while many works only give asymptotical analysis.

+ Explain why finite CNNs outperforms infinite CNNs and quantify the difference.

Cons:

- The presentation can be improved. For example, Sections 2 and 3 would better if the contents can be further organized into smaller parts.

Questions:

* How different is the techniques used in the proves comparing to existing results? I have gone through the proves and the techniques wherein seem similar to existing works for me. Could you please give detailed proof skeletons in the manuscript and discussion the differences with related works?

---

> ### Author Response · Authors · 2020-11-15
> **Response to Reviewer 3 - Difference in the derivation of the results**
>
> Dear reviewer,
>
> Thank you for your review.
> The detailed derivations for the FWC can be found in appendices A-C.
>
> Could you be more specific regarding your question about the difference in the techniques used compared to existing results?
> Are you referring to Yaida 2020? Or to other previous works?
> We listed the main differences in our work from previous work in the "Related work" section.
>
> Regarding Yaida 2020, there are indeed some similarities, as both works provide the FWC to the predictive mean in terms of the 4th cumulant of the distribution of the DNN output. However, there are several important differences in terms of the technique and the resulting analytical expression:
> 1) As we mentioned in the paper, we represent a distribution over *function space* for arbitrary training and test data, rather than over specific draws of data sets, as done in Yaida 2020. This is useful for studying the large $n$ behavior of learning curves, where analytical insights into generalization can be gained
> 2) In Yaida 2020 the expression contains a summation over both train and test indices so that if we were to receive additional test points we would need to recompute, whereas our result only has a summation over the training set, which is also computationally cheaper.
> 3) Our expressions for the predictive mean are arguably simpler and we offer an intuitive interpretation of them in terms of the discrepancy of the GP predictions.
> 4) We also provide an expression for the FWC for the predictive variance at each test point (2nd lines in Eqs. 10-12).
>
> If you have further questions regarding this point or other points, we would be happy to answer.

---

### Official Review · AnonReviewer2 · 2020-10-29
**The presentation is very ambiguous. Overclaimed the implications.**

**Rating:** 5
**Confidence:** 3

**Review:**

This paper shows a correspondence between deep neural networks (DNN) trained with noisy gradients and NNGP. It provides a general analytical form for the finite width correction (FWC) for NNSP expanding around NNGP. Finally, it argues that this FWC can be used to explain why finite width CNNs can improve the performance relative to their GP counterparts on image classification tasks.

This paper is written in a physics style which makes it hard to read. Many notations are defined in words instead of mathematical formulas. This is quite annoying since it is hard to follow what are scalers, vectors, matrices, functions, random variables, and what are the arguments of the functions.
For the first claimed contribution, I feel that this is a simple observation rather than a significant contribution: the authors just pointed out that the stationery distribution of Langevin dynamics is a Gibbs measure.
For the third claimed contribution, I feel that the authors over-claimed what their results can suggest. The authors showed that the FWC contains the information of the weight sharing structure in CNN, and then cited a 'common practice' to argue that this weight sharing structure will lead to better test error. This doesn't give a convincing argument that finite width CNN will outperform its corresponding GPs.

The derivation of the finite width correction of the NNGP could indeed be a concrete contribution, but the result is presented in a very ambiguous way. For example, the $U_{x1, x2, x3, x4}$ quantity seems to be a very important quantity in defining this finite width correction. However, its definition in Eq. (9) is quite ambiguous. Especially, U_{x1, x2, x3, x4} depend on $\phi_\alpha = \phi(z_i^{l-1}(x_\alpha))$, and I didn't see where does the subscript $i$ come from. The authors used the Einstein convention in subscripts but I don't know what are ranges of \alpha, \beta, \gamma, and \delta to be summed over (is it 1 to n or 1 to 4?). There are many other places where the quantities are not concretely defined and I can only guess what do these notations mean in order to follow the results.

Although the finite width correction of NNGP (if properly written) could be an interesting result, due to the ambiguous writing style of the paper, I don't think it achieves the goal of clearly presenting the results. I will suggest a rejection.

After reading the rebuttal:
I increased my score to 5. I still feel that the writing style is hard to follow. Besides the examples that I wrote in my original review, there are many other places where the notations and definitions are not clearly written.

---

> ### Author Response · Authors · 2020-11-24
> **Response to Reviewer 2 - part 1**
>
> Thank you for your review. We will answer your questions in order of appearance:
>
> 1. **"Physics style" writing** -
> We agree that this paper is written in a "physics style", i.e. we did not attempt to write using strict mathematical rigor of definitions and proofs, though we believe our analytical results are correct. Much like physics, deep learning is an experimentally driven field. In such circumstances, we believe the physics-style paradigm of research is both appropriate and necessary. Had we limit our theories in physics to those we can prove using lemmas and strict definitions, physics with its many successes would have been greatly diminished. We believe that the CS community would in the long term benefit from hosting this paradigm of research. Notwithstanding we accept that the definitions of certain objects could be better specified and will correct this in the revised version.
>
> 2. **First claimed contribution (NNGP correspondence in the $N\to\infty$ limit for trained DNNs)** -
> the claim is non-trivial, and does not simply amount to the stationary distribution of Langevin dynamics being a Gibbs measure, for the following two reasons:
>
> (a) **Weight space versus function space**:
> The transition between Eqs 2 and 3 gives the distribution in function space, not weight space. It shows that $P[f]$ is a product of a data term that is independent of the NN architecture $e^{-\mathcal{L}[f]/2\sigma^{2}}$
> and the prior term $P_0[f] = \int dw e^{-\frac{1}{2\sigma_w^2}||w||^2}\delta[f - z_w]$
> which does depend on the architecture and for which we perform the Edgeworth expansion to obtain the FWC. To compare this to some related results:
>
> i. Convergence to the NNGP was shown by Cho \& Saul 2009, but importantly in that work the weights were simply drawn iid at initialization. In comparison, in our paper the weights are sampled across the noisy training dynamics, and given the NN's architecture, this gives rise to an empirical distribution over the NN's output, namely a distribution over function space.
>
> ii. Using a Langevin dynamics argument one can arrive at a distribution in weight space which would be highly non-Gaussian due to the data-term. By using our path-integral approach and re-phrasing this in function space the distribution becomes Gaussian (at $N \to \infty$).
>
> (b) **Ergodicity of dynamics in function space in a reasonable time** -
> Our numerical experiments provide evidence that such Langevin dynamics of the weights yield valid samples from the posterior over functions $P[f]$, with reasonable (not exponential) training times. This is also highly non-trivial, and to the best of our knowledge not stated previously in the literature.
>
> 3. **Third claimed contribution (FWCs can improve the performance of finite CNNs)** -
>
> (a) It is well known in the literature that finite width CNNs generally outperform their respective GPs on image classification tasks (Novak et al. 2018, Arora et al. 2019). It remains unclear whether this is due to the finite width itself or to other details of the training procedure that distinguish finite CNNs from their respective GPs, such as finite learning rates, early stopping etc (Lee et al. 2020).
>
> (b) The solid green line in figure 3 demonstrates, at least under the settings we used, that the performance of a finite width CNN tends to that of the corresponding GP, and that it outperforms it for some finite width (i.e. number of channels). This suggests that the difference in performance between the finite width CNN and its corresponding GP is a result of the finite width itself and not the aforementioned factors.
>
> (c) We accept that the argument could be made more convincing by considering a more controlled toy-example, that better illustrates under what specific conditions the FWC will improve performance. We are currently working on a follow-up paper that does this.

---

> > ### Author Response · Authors · 2020-11-24
> > **Response to Reviewer 2 - part 2**
> >
> > (4) **Derivation of the FWC result** -
> >
> > (a) Our derivation rests on the Edgeworth expansion, which gives the corrections to the Central Limit Theorem for a **finite** sum of iid RVs in terms of cumulants, which for the multivariate case become tensors. In our case, we are considering random functions defined by our stochastic process (the NNSP), thus the cumulants are functional tensors, i.e. are continuously indexed by the inputs $x_\alpha$. For example, the second order cumulant is simply the covariance function (i.e. kernel) $K\left(x_{1},x_{2}\right)$ and the fourth order cumulant is $U\left(x_{1},x_{2},x_{3},x_{4}\right)$. We will include a short recap of this in the revised version which will hopefully clarify this. As mentioned in the paper, since the NN weights are centered, all odd cumulants vanish and thus, conveniently, the fourth-order cumulant $U$ is given in terms of the fourth moment and second moment, with the three distinct permutations coming from Wick's theorem, as in Eq. 9.
> >
> > (b) The subscript $i$ is the neuron index, it does not affect the value of $U$, thus it is omitted for brevity.
> > Just as for the kernel we could write
> > $K^{\left(\ell\right)}\left(x_{\alpha_{1}},x_{\alpha_{2}}\right)=\left\langle \phi_{\alpha_{1}}^{\left(\ell-1\right)}\phi_{\alpha_{2}}^{\left(\ell-1\right)}\right\rangle $
> > where
> > $
> > \phi^{\left(\ell-1\right)}\left(z_{i}\left(x_{\alpha_{1}}\right)\right):=\phi_{\alpha_{1}}^{\left(\ell-1\right)}
> > $
> > and this is the same, regardless of the neuron index $i$.
> >
> > (c) The indices in Eq. (9) range from $1$ to $4$ (here there is no Einstein summation, as every index appears exactly once in each term), and the indices in Eq. (11)-(12) range from $1$ to $n$, as stated clearly in the line between these equations.
> >
> > (d) It would be helpful if you could specify which other notations you found unclear.

---

### Official Review · AnonReviewer4 · 2020-11-02
**An interesting paper but has some flaws**

**Rating:** 5
**Confidence:** 5

**Review:**

 In this paper, the authors propose an analytical framework for DNN training with noisy gradient descent and weight decay. For infinite width, the authors have established a correspondence between DNNs trained with
noisy gradients and the NNGP, and for finite width, the authors introduce a finite width correction (FWC). Overall, the paper is quite interesting and well organized. However, I still have several concerns.

While I agree with the authors that this is an analytical framework different from the Neural Tangent Kernel, my major concern is that this paper doesn't show the advantage of the new framework because this paper hasn't provided any theoretical result for convergence or generalization.

The authors have commented in the introduction that for NTK "deterministic training is qualitatively different from the stochastic one used in practice, which may lead to poorer performance when combined with a small learning rate." However, as far as I know, there are plenty of good works that have considered NTK trained with SGD([1], [2], [3]). The authors may want to comment on these papers. In the infinite width limit, the authors establish a correspondence between DNNs trained with noisy gradients and the NNGP. Actually, there is a line of work that directly studies the infinite wide NN trained with noisy gradients and weight decay regularization in the so-called mean-field regime([4], [5], [6]).  It would be much better if the authors can characterize the difference between NNGP and those papers I have mentioned.

My other minor concerns are as follows:

1. In the eq 2 of this paper, it seems that the probability measure of the parameter is very close to the gaussian with \sigma_{w}^2 variance. Would it be better for the NN to initialize at \sigma_{w}^2 too? If so, whether the parameters will still only make a small change in the distribution space?

2. The analysis in this paper highly depends on the ergodicity of the dynamics. The authors mention that a small ACT of the dynamics implies this key property, especially for the output. It would be better if the authors can explain more about why the ACT can imply ergodicity.
3. There is no explicit theorem, lemma, or proof in this paper, which is quite hard to read or follow. I suppose the finite width correction would be the key theorem of this paper?

References:

[1] Allen-Zhu, Z., Li, Y., & Song, Z. (2019, May). A convergence theory for deep learning via over-parameterization. In International Conference on Machine Learning (pp. 242-252). PMLR.

[2] Zou, D., Cao, Y., Zhou, D., & Gu, Q. (2020). Gradient descent optimizes over-parameterized deep ReLU networks. Machine Learning, 109(3), 467-492.

[3] Daniely, A. (2017). SGD learns the conjugate kernel class of the network. In Advances in Neural Information Processing Systems (pp. 2422-2430).

[4] Mei, S., Misiakiewicz, T., & Montanari, A. (2019). Mean-field theory of two-layers neural networks: dimension-free bounds and kernel limit. arXiv preprint arXiv:1902.06015.

[5] Chen, Z., Cao, Y., Gu, Q., & Zhang, T. (2020). Mean-Field Analysis of Two-Layer Neural Networks: Non-Asymptotic Rates and Generalization Bounds. arXiv preprint arXiv:2002.04026.

[6] Tzen, B., & Raginsky, M. (2020). A mean-field theory of lazy training in two-layer neural nets: entropic regularization and controlled McKean-Vlasov dynamics. arXiv preprint arXiv:2002.01987.

---

> ### Author Response · Authors · 2020-11-24
> **Response to Reviewer 4 - part 1**
>
> 1. **Advantage of the new framework (wrt NTK) and theoretical result for convergence or generalization**-
> We believe that our framework is important and that it has several advantages wrt NTK:
>
> (a) **Valid under different settings** - Our framework is a complementing one to NTK, valid under different settings: noisy gradients and weight decay, which makes it qualitatively different from NTK.
>
> (b) **Convergence** - Due to the noisy gradients and weight decay, our training protocol does not converge to a global optimum of the training loss, or in fact to any specific point in weight space, since the NN weights continue to fluctuate as the training dynamics are run. Instead, it is more appropriate to ask whether the dynamics generate valid samples from the posterior distribution ($P[f]$) after a reasonable amount of epochs. We address this below in item 5 on ergodicity.
>
> (c) **Generalization** - We provided a mapping to the Equivalent Kernel (EK), which is a good approximation in the large training set regime, which gives an analytical result for training curves, thus quantifying generalization properties.
>
> (d) **Simpler and more interpretable FWC expressions** -
> In the paper, we quantify how the outputs of a finite DNN deviate from a GP. We provide much simpler and more interpretable analytical expressions for the FWC, relative to the ones derived for the NTK (see Eq 13 in Dyer \& Gur-Ari 2019). Also, unlike Dyer \& Gur-Ari 2019 we provide FWCs on a generic data point and not just on the training set.
>
> (e) **Regime of validity of GP correspondence** -
> Our framework indicates under what conditions (NN width, architecture, hyper-parameters, number of examples) the GP correspondence is a good approximation to the NN output in practice. Specifically, one of the interesting conclusions of our derivation is that the DNN converges more rapidly to its $N\rightarrow \infty$ limit when its corresponding GP has a small error.
>
> 2. **NTK trained with SGD** - The authors of Refs. [1],[2],[3] that you mentioned indeed consider over-parameterized NNs trained with SGD (as well as full batch GD). However, [1],[2] do not discuss at all any correspondence to the NTK or other kernel methods. Also, they all generally require small learning rates for their convergence proofs and do not use weight decay in the training procedure. Our paper is different from these works in several regards:
>
> (a) **SGD vs GD with injected white noise** -
> In our setting, state-independent white noise is injected at every time step, unlike in SGD which has state-dependent noise, e.g. if SGD dynamics reach zero training error, the gradients will vanish and the dynamics will come to a fixed point, whereas for injected white noise this can never happen and the weight will continue to fluctuate indefinitely.
>
> (b) **Gradient flow vs. SGD with small learning rates** -
> We expect that there will not be any qualitative difference between SGD with sufficiently small learning rates and deterministic gradient flow (i.e. full-batch GD with vanishing learning rate) which was used in the original NTK setting. Thus we believe our statement that you cited holds true even if gradient flow is replaced by SGD with sufficiently small learning rates.
>
> 3. **Training with noisy gradients and weight decay regularization in the mean-field regime** -
> Thank you for pointing out these references, we agree that they constitute an interesting and relevant line of work and will cite them in the revised version, as well as point out the main differences from our paper. We list these differences here as well:
>
> (a) **Different scaling** -
> These papers consider the mean field regime in which the NN output has the following scaling with width:
>     $f\left(x;W,a\right)=\frac{1}{N}\sum_{j=1}^{N}a_{j}\phi\left(w^{j}\cdot x\right)$
>     whereas in our paper we considered the "kernel regime" with the following scaling with width:
>     $f\left(x;W,a\right)=\frac{1}{\sqrt{N}}\sum_{j=1}^{N}a_{j}\phi\left(w^{j}\cdot x\right)$. This difference in scaling yields very different models.
>
> (b) **Function space vs. distribution space** -
> In the mean-field regime, one is interested in the dynamics (finite $t$) of the distribution over the NN parameters in the form of a PDE of the Fokker-Planck type. In contrast, in our framework, we are interested in the distribution over function space at equilibrium, i.e. for $t\to\infty$.
>
> (c) **Adequacy for shallow vs. deep nets and fully-connected vs CNN** -
> It seems that all three papers you cited only consider two-layer fully-connected NNs and that the mean-field analysis they carry out is tailored for this case and is hard to generalize to deeper nets or to CNNs. In contrast, our formalism generalizes to deeper fully-connected NNs and to CNNs as well, as we showed in section 4.2.

---

> > ### Author Response · Authors · 2020-11-24
> > **Response to Reviewer 4 - part 2**
> >
> > 4. **Eq 2, weights initialization and changes in weights distribution** -
> > Eq. 2 refers to the probability distribution of the parameters at the end of training, i.e. it is the posterior distribution. Strictly speaking, it will be close to a Gaussian with variance $\sigma_w^2$ only if the data term is small namely
> > $\frac{1}{2\sigma_{w}^{2}} ||w||^2 \gg \frac{1}{2\sigma^{2}} \mathcal{L} \left(z_{w}\right) $.
> > This can happen if there are very few training points or if we choose the hyper-parameters of the training protocol such that
> > $\sigma_{w}^{2} \ll \sigma^{2}$
> > namely very strong weight decay $\gamma \gg 1$. While we haven't explored this in detail we conjecture that for large enough $N$ the distribution for each individual weight would indeed be close to Gaussian. It is only their cumulative effect on $f(x)$ which changes drastically during learning.
> >
> > (a) **Initialize the NN to $\sigma_w^2$** -
> > you are correct in noticing that $\sigma_w^2$ refers to the training dynamics and this could in principle be different from the variance of the weights at initialization. However, in practice, we did in fact initialize the weights with variance $\sigma_w^2$. This choice does not affect our claims but we believe it is a good choice since it will make the dynamics converge to the equilibrium distribution in minimal training time.
> >
> > (b) **Small changes in the distribution of weights?** -
> > You raise an interesting question. Unlike in Ref. [5], in our setting there will generally be \textbf{large} changes to the weights and potentially also large changes to their distributions. This is another qualitative difference from the NTK and its generalization to the parameter distribution space described in [5]. This is due to the ergodicity of the dynamics which "forget" the initial condition. Again, this is true provided that the data term is not very small.

---

> > > ### Author Response · Authors · 2020-11-24
> > > **Response to Reviewer 4 - part 3**
> > >
> > > 5. **Ergodicity and ACT** -
> > > This is indeed a subtle and important point, and we thank you for pointing this out. We will revise this section to be more precise.
> > >
> > > (a) **Self-consistency argument** -
> > > As we mentioned in the paper, ergodicity is in general hard to prove rigorously and thus we must revert to heuristics. We believe the most robust evidence of ergodicity in function space is the high level of accuracy of our analytical expressions wrt to our numerical results. This is a self-consistency argument: we assume ergodicity in order to derive our analytical results and then indeed find that they agree very well with the experiment, thus validating our original assumption.
> > >
> > > (b) **ACT and ergodicity** -
> > > Short ACT does not logically imply ergodicity, we only mentioned it as an "indicator" of ergodicity. One could indeed construct examples of a loss landscape where noisy dynamics travel only in some local energy valley, thus not sampling the entire space, but with small ACT. In fact, the converse is true: exponentially long ACT implies non-ergodic dynamics. However, the empirical ACT gives a lower bound on the true correlation time of the dynamics.
> > >
> > > (c) **Ergodicity in the mean** -
> > > In fact, full ergodicity may be too strong of a condition and we don't really need it for our purposes, since we are mainly interested in collecting statistics that will allow us to accurately compute the posterior mean of the distribution in function space. Thus, a weaker condition that is sufficient here is **ergodicity in the mean**, and we believe our self-consistent argument above demonstrates that it holds. This was mentioned before in the appendix, but now we have moved it to the main text in the revised version.
> > >
> > > 6. **Key result and writing style** -
> > > This paper is purposely not written in the format of "definition, lemma, theorem, proof". Instead, we adopt a "physics writing style" where the rigorous definitions, statements, and proofs are replaced by derivations that rest on some reasonable assumptions, that we later validate numerically. We hope that the revised version will be easier to follow.
> > > One could say that the finite width correction is one of our key analytical results, but as stated in the abstract, it is not the only one:
> > >
> > > (a) **NNGP correspondence from gradient-based training** -
> > > We believe that the convergence in our setting to NNGP as $N\to\infty$ was not previously recognized. Previous works on the NNGP, e.g. Lee at al. 2017, considered the correspondence where the weights are sampled \textit{at initialization} to give rise to the GP prior, and then Bayesian inference is carried out, without any reference to gradient-based training. In contrast, in our setting the distribution of the weights comes about from the noisy gradient descent dynamics, so that the weights drift far away from their initial values.
> > >
> > > (b) **Characterizing the behavior of the FWC with data set size $n$** -
> > > see section 3.2, and especially Eq 14 and Fig 1. Interestingly, we found that the FWC can be written explicitly in terms of the discrepancy of GP in predicting the target $g$ and the 4th cumulant $U_{x_1, x_2, x_3, x_4}$.

---

### Author Response · Authors · 2020-11-24
**General response to all reviewers**

To all reviewers,

We thank you for the time you spent reading our paper and your helpful feedback. We feel that most of the comments concern presentation style, while most of you agree that this paper addresses an important and interesting question with interesting results. We hope that the revised version addresses your concerns and that you will consider raising your scores.

---

### Decision · Program_Chairs · 2021-01-07
**Final Decision**

**Decision:**

Reject

**Comment:**

This paper develops an interesting new angle on the behavior of large-width neural networks by elucidating the connection between the NNGP and noisy gradient descent and by examining finite-width corrections through an Edgeworth expansion. While these contributions are important, the paper would better serve the community if its presentation were significantly improved before publication. The main issue is not one of presentation style -- papers with physics-style prose are welcomed and appreciated at ICLR -- but rather one of presentation substance. In addition to the various specific points raised by the reviewers, I would add that the figures and captions are difficult to interpret, the experiments need a more in-depth discussion, and the notations should all be defined at the time of their introduction, among other things. For these reasons, I cannot recommend accepting the paper in its current form, but I hope to see a more polished version of the manuscript at a subsequent conference.